# Alternative splicing controls teneurin-latrophilin interaction and synapse specificity by a shape-shifting mechanism

Jingxian Li[1,2,5], Yuan Xie [1,5], Shaleeka Cornelius[3,4], Xian Jiang[3,4], Richard Sando[3,4], Szymon P. Kordon [1,2], Man Pan[1], Katherine Leon[1,2], Thomas C. Südhof [3,4], Minglei Zhao [1✉] & Demet Araç [1,2✉]

The trans-synaptic interaction of the cell-adhesion molecules teneurins (TENs) with latrophilins (LPHNs/ADGRLs) promotes excitatory synapse formation when LPHNs simultaneously interact with FLRTs. Insertion of a short alternatively-spliced region within TENs abolishes the TEN-LPHN interaction and switches TEN function to specify inhibitory synapses. How alternative-splicing regulates TEN-LPHN interaction remains unclear. Here, we report the 2.9 Å resolution cryo-EM structure of the TEN2-LPHN3 complex, and describe the trimeric TEN2-LPHN3-FLRT3 complex. The structure reveals that the N-terminal lectin domain of LPHN3 binds to the TEN2 barrel at a site far away from the alternatively spliced region. Alternative-splicing regulates the TEN2-LPHN3 interaction by hindering access to the LPHN-binding surface rather than altering it. Strikingly, mutagenesis of the LPHN-binding surface of TEN2 abolishes the LPHN3 interaction and impairs excitatory but not inhibitory synapse formation. These results suggest that a multi-level coincident binding mechanism mediated by a cryptic adhesion complex between TENs and LPHNs regulates synapse specificity.

---

[1] Department of Biochemistry and Molecular Biology, The University of Chicago, Chicago, IL 60637, USA. [2] Grossman Institute for Neuroscience, Quantitative Biology and Human Behavior, The University of Chicago, Chicago, IL 60637, USA. [3] Department of Molecular and Cellular Physiology, Stanford University, Stanford, CA 94305, USA. [4] Howard Hughes Medical Institute, Chevy Chase, MD, USA. [5]These authors contributed equally: Jingxian Li, Yuan Xie. ✉email: mlzhao@uchicago.edu; arac@uchicago.edu

Neural circuit assembly and function in the central nervous system requires precise formation and specification of diverse excitatory and inhibitory synapse subtypes. Imbalances in the ratio of excitatory to inhibitory synapse function are thought to be a major component of brain disorders such as autism, mental retardation, and attention deficit hyperactivity disorder (ADHD)[1]. Recent work suggested that combinatorial sets of trans-synaptic interactions between cell-adhesion molecules, including teneurins (TENs or ODZs) and latrophilins (LPHNs or ADGRLs), mediate synapse formation and regulate the exquisite specification of synapses, but the underlying molecular mechanisms remain largely unexplored[2,3].

TENs and LPHNs are evolutionarily conserved cell-surface proteins. While the roles of TENs and LPHNs in early organisms remain unclear, they have critical roles in embryonic development and brain wiring in higher eukaryotes. TENs (TEN1-4 in mammals) are large type-II transmembrane proteins that are composed of an N-terminal cytoplasmic sequence, a single transmembrane region, and a large extracellular region (ECR) composed of >2000 amino acids with partial homology to bacterial Tc toxins (Fig. 1a)[4–7]. They form constitutive *cis*-dimers via highly conserved disulfide bonds formed in proximity to their transmembrane helix (Fig. 1c)[7–9]. TENs have central roles in tissue polarity, embryogenesis, heart development, axon guidance, and synapse formation[6,10–17]; and are linked to various diseases including neurological disorders, developmental problems, various cancers, and congenital general anosmia[18–24]. LPHNs (LPHN1-3 in mammals) belong to the adhesion-type G-protein coupled receptor (GPCR) family[25–27] and have a large N-terminal ECR (>800 amino acids) in addition to their signaling seven-pass transmembrane domain and cytoplasmic tail (Fig. 1b)[26,28–31]. LPHNs have roles in embryogenesis, tissue polarity, synaptic development, and neural circuit connectivity, interestingly almost identical to the functions of TENs[15,16,26,28–33] (Fig. 1b). LPHN3 mutations are linked to ADHD, as well as numerous cancers in humans[34–36].

The large ECRs of TENs and LPHNs form a tight trans-cellular adhesion complex[6,15,16]. The most N-terminal Lectin (Lec) and Olfactomedin (Olf) domains of LPHN interact with TEN2, with the Lec domain contributing most of the binding affinity (Fig. 1b). A four or five amino acid splice insert (MEQK or KVEQK) between these domains decreases the affinity of the TEN/LPHN interaction[16]. In addition to TENs, LPHNs form trans-cellular interactions also with homodimeric cell-adhesion molecules called fibronectin leucine rich repeat transmembrane proteins (FLRTs), which further interact with Uncoordinated5 (UNC5s)[27,37]. The LPHN3/TEN2 interaction as well as the LPHN3/FLRT3 interaction were individually reported to be important for synapse formation and organization[15,16,37]. Recent work showed that in transgenic mice in vivo, postsynaptic LPHN3 promotes excitatory synapse formation by simultaneously binding to TEN and FLRT, two unrelated presynaptic ligands, which is required for formation of synaptic inputs at specific dendritic localizations[3]. Conversely, LPHN3 deletion had no effect on inhibitory synapse formation[3]. The precise molecular mechanisms of neither excitatory nor inhibitory synapse formation are known.

In addition to the critical role of coincident TEN2 and FLRT3 binding to LPHN3 for specification of excitatory synapses, alternative splicing of TEN2 also plays a crucial role in specifying excitatory vs. inhibitory synapses[2]. An alternatively spliced seven-residue region (NKEFKHS) within TEN2 acts as a switch to regulate trans-cellular adhesion of TEN2 with LPHNs and to induce different types of synapses in vitro[2]. The TEN2 −SS splice variant that lacks the splice insert can bind to LPHN3 in *trans* (Fig. 1c, left side)[2]. However, TEN2 +SS, the splice variant that includes the seven amino acids is unable to interact with LPHN3 in *trans* in identical experiments (Fig. 1c, right side)[2]. Similarly, the same alternatively spliced site also may regulate TEN2 trans-homodimerization[13], although no such trans-homodimerization could be detected in some assays[16]. However, the molecular mechanism of how alternative splicing regulates ligand interactions is unclear.

In agreement with in vivo transgenic mice experiments, only the splice variant of TEN2 that can interact with LPHN3 (TEN2 −SS) was able to promote excitatory synapses when co-expressed with FLRT3 in cultured neurons in vitro[3]. The other TEN2 splice variant that cannot interact with LPHN3 (TEN2 +SS) did not promote excitatory synapse formation when expressed alone or co-expressed with FLRT3. Instead, this TEN2 isoform induced inhibitory postsynaptic specifications in a LPHN-independent manner likely by interacting with other unknown ligands, suggesting that alternative splicing of TEN2 regulates excitatory vs. inhibitory synapse specification[2]. These results indicate a multi-level coincidence signaling mechanism for the specification of synaptic connections that requires the presence of the proper combination of molecules and their appropriate alternatively spliced isoforms to colocalize in order to induce the formation of a specific type of synapse. However, the molecular details of the TEN/LPHN/FLRT interaction are not known. Furthermore, the structural basis for the lack of the TEN2/LPHN3 *trans* interaction in the presence of a short splice insert in TEN2 is unclear.

Here, we have determined the 2.9 Å resolution cryo-EM structure of the TEN2/LPHN3 complex and described the direct and simultaneous interaction of LPHN3 with both TEN2 and FLRT3. The TEN2/LPHN3 complex structure revealed that the N-terminal Lec domain of LPHN3 interacts with the β-barrel domain of TEN2. Both the Lec and the preceding Olf domains of LPHN3 face away from the alternatively spliced site within the β-propeller domain of TEN2, indeed providing no explanation for how the short splice insert may regulate TEN2/LPHN3 interaction. Using a series of experimental setups that mimic either *trans*-cellular interactions between opposing cell-membranes, or *cis-like*-interactions in solution, we showed that alternative splicing of TEN2 indirectly regulates the TEN2/LPHN3 interaction by altering the accessibility of the LPHN-binding site on TEN2 with the help of membranes, rather than directly interfering with the LPHN-binding site. Mutagenesis of the LPHN-binding site on TEN2 abolished the TEN2/LPHN3 interaction and had a severe and specific effect on excitatory synapse formation, but had no effect on inhibitory synapse formation. These results provide a molecular and mechanistic understanding of the multi-level coincidence binding mechanism that mediates specificity in synapse formation and circuit-wiring.

## Results

**Structure of the TEN2/LPHN3 complex**. To determine the structure of the TEN2/LPHN3 complex, the ECR of human TEN2 lacking the EGF repeats that are responsible for *cis*-dimerization (TEN2 −SS ECRΔ1, encoding residues 727-2648, Fig. 1a) and the full ECR of human LPHN3 (LPHN3 +SS ECR, encoding residues S21-V866, Fig. 1b) were co-expressed. The complex structure was determined by single-particle cryo-EM. After multiple rounds of 3D classification, two cryo-EM maps were obtained: one corresponding to the monomeric TEN2 −SS ECRΔ1 in complex with LPHN3 ECR at a nominal resolution of 2.9 Å (from 9.7% of particles, Supplementary Figs. 1–4), and the other corresponding to the monomeric TEN2 ECRΔ1 with a better resolved β-propeller domain at a nominal resolution of 3.0 Å (from 3.8% of particles). A near-atomic resolution model of the protein complex was built using the available TEN2 structure

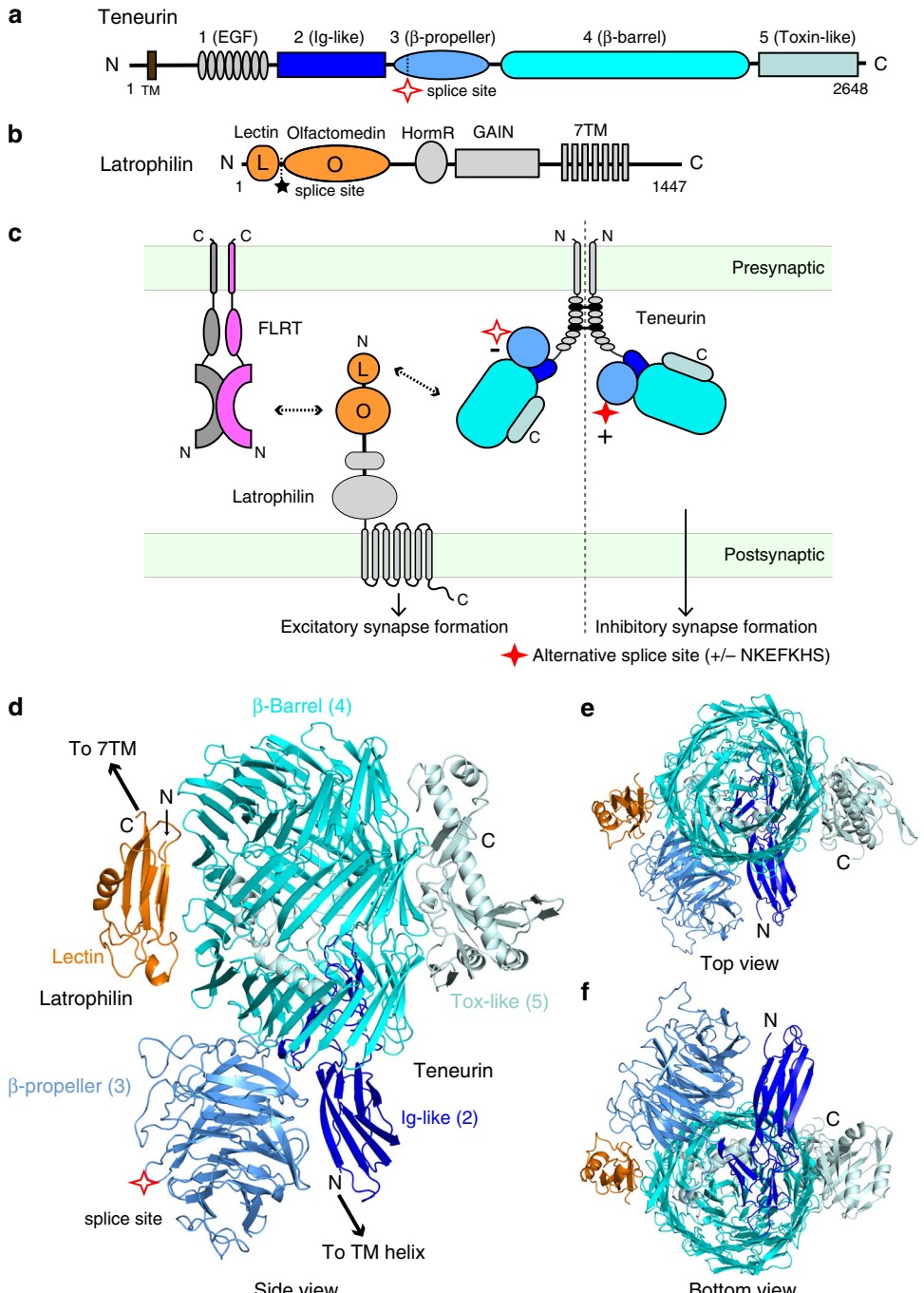

**Fig. 1 The structure of the TEN2/LPHN3 complex. a** Schematic diagram of human TEN2. Extracellular domains are colored gray, dark blue, sky blue, cyan, and palecyan for domains 1–5, respectively; transmembrane region (TM) in brown. Domain numbers and descriptions are indicated above scheme. **b** Schematic diagram of human LPHN3. Lec and Olf domain are colored orange; Hormone Receptor (HormR), GAIN and TM domains are colored gray. Splice site of TEN2 and LPHN3 are shown as red star and black stars, respectively. The constructs that were used in structure determination were TEN2 −SS and LPHN3 +SS. **c** Schematic diagram of the interaction network between TEN, LPHN, and FLRT at the synapse. TEN and FLRT are localized on the presynaptic cell membrane, while LPHN is localized on the postsynaptic membrane. TEN2 −SS isoform (empty red star) forms trans-cellular complexes with LPHN3 and induces excitatory postsynaptic specializations when LPHN3 simultaneously binds to FLRT3 (left). In contrast, TEN2 +SS isoform (filled red star) induces inhibitory postsynaptic specializations independent of TEN2/LPHN3 interaction or FLRT3 binding (right). **d-f** The structure of the TEN2/LPHN3 complex as obtained from single-particle cryo-EM analysis. The Lec domain of LPHN3 binds to the side of the β-barrel of TEN. The alternatively spliced site within the β-propeller domain is distal to the Lec binding site on TEN.

(PDB: 6CMX) and the Lec Olf structure (PDB:5AFB) (Fig. 1d, Supplementary Figs. 1–4, Table 1).

Our TEN2/LPHN3 complex structure comprises a heterodimer of ~100 × 50 × 115 Å in which the Lec domain of LPHN3 interacts with the side of the barrel domain of TEN2 (Fig. 1d±f).

The TEN2 ECR is assembled as a large cylindrical barrel sealed by the β-propeller and Ig-like domains at the bottom; and the toxin-like domain protrudes from and attaches to the side of the barrel as previously reported[2,38]. The Lec domain of LPHN3 and the toxin-like domain of TEN2 bind to opposite faces of the β-

**Table 1 Cryo-EM data collection, refinement and validation statistics.**

| | #1 TEN2/LPHN3 complex (EMD-21205) (PDB 6VHH) | #2 TEN2 (EMD-21205) |
|---|---|---|
| Data collection and processing | | |
| Magnification | 81,000 | 81,000 |
| Voltage (kV) | 300 | 300 |
| Electron exposure (e–/Å$^2$) | 60.1 | 60.1 |
| Defocus range (μm) | −1.0 to −2.5 | −1.0 to −2.5 |
| Pixel size (Å) | 0.54 | 0.54 |
| Symmetry imposed | C1 | C1 |
| Initial particle images (no.) | 4,475,958 | 4,475,958 |
| Final particle images (no.) | 436,208 | 170,133 |
| Map resolution (Å) | 2.97 | 3.07 |
| FSC threshold | 0.143 | 0.143 |
| Map resolution range (Å) | 2.4-4.5 | 2.4-4.5 |
| Refinement | | |
| Initial model used (PDB code) | 6CMX | |
| Model resolution (Å) | 3.0 | |
| FSC threshold | 0.5 | |
| Model resolution range (Å) | | |
| Map sharpening $B$ factor (Å$^2$) | | |
| Model composition | | |
| Non-hydrogen atoms | 14271 | |
| Protein residues | 1821 | |
| Ligands | BMA:5 | |
| | NAG:14 | |
| $B$ factors (Å$^2$) | | |
| Protein | 53 | |
| Ligand | 67 | |
| R.m.s. deviations | | |
| Bond lengths (Å) | 0.015 | |
| Bond angles (°) | 1.034 | |
| Validation | | |
| MolProbity score | 2.02 | |
| Clashscore | 10.16 | |
| Poor rotamers (%) | 0.61 | |
| Ramachandran plot | | |
| Favored (%) | 91.78 | |
| Allowed (%) | 8.05 | |
| Disallowed (%) | 0.17 | |

barrel in a seemingly parallel orientation to each other (Fig. 1d–f). No major conformational changes are observed when the complex structure is compared with the individual structures of TEN2 or LPHN3.

Analysis of the cryo-EM maps at a lower threshold also revealed continuous density for the Olf domain that extended from the Lec domain towards the opposite side from the β-propeller domain of TEN2 (Fig. 2a). In spite of the lower resolution, it was possible to fit the available LPHN3 Olf domain structure in this density (Fig. 2b). The Olf domain is positioned in close proximity to the top of the TEN2 β-barrel, although it is not in contact with TEN2 (Fig. 2b). The presence of the splice insert (KVEQK) between the Lec and OLF domains in our LPHN3 construct likely causes the lack of interaction of Olf domain with TEN. The remaining C-terminal domains of LPHN3 for5 which there is no EM density likely extend from the opposite side, away from the TEN2 TM domain located at the TEN2 N-terminus. This orientation positions the membrane-anchored TM domain of LPHN3 on the opposite side from the membrane-anchored TM domain of TEN, and thus is compatible with a *trans*-cellular interaction of TEN with LPHN (Fig. 2c).

**LPHN3, TEN2, and FLRT3 form a trimeric complex**. LPHN proteins are involved in heterodimeric interactions with TENs and FLRTs and coincidence binding of both FLRTs and TENs is required for excitatory synapse formation (Fig. 1c, left). Additionally, FLRTs interact with UNC5s and form homodimers that are incompatible with their UNC5 binding. However, whether these interactions are compatible is unclear. Thus, we investigated whether the TEN2, LPHN3, and FLRT3 interactions are compatible with each other, or in other words, whether TEN2, LPHN3, and FLRT3 can form a trimeric complex. The availability of the LPHN3/FLRT3 complex structures and of the LPHN3 Lec–Olf structure enabled us to compare structures, and to

predict and test the compatibility of the possible interactions of TEN, LPHN3, and FLRT3[39]. Intriguingly, superimposition of the Lec domain from the LPHN3/FLRT3 structure with the Lec domain from the LPHN3/TEN2 complex structure showed that FLRT3 and TEN2 bind to distinct domains on LPHN3 and that there are no clashes between TEN2 and FLRT3, suggesting that LPHN3 can simultaneously bind to TEN2 and FLRT3 (Fig. 2c, d).

In order to test whether this model is correct, we co-expressed FLRT3 LRR, LPHN3 full ECR, and TEN2 −SS full ECR, purified the complex by gel filtration chromatography and analyzed the fractions by SDS-PAGE (Fig. 2e). Both LPHN3 ECR and FLRT3 LRR elution volumes shifted to the left as compared to the elution volumes of the individual proteins. All three proteins eluted in the same fractions, indicating the formation of a trimeric TEN2/LPHN3/FLRT3 complex (Fig. 2e). These results suggest that TEN2 and FLRT3, both ligands of LPHN3, can simultaneously bind to LPHN3 and form a trimeric complex in vitro, supporting the in vivo observations that coincident binding of both TEN2 and FLRT3 to LPHN3 is required for excitatory synapse formation[3].

**The binding interface of the TEN2/LPHN3 complex is conserved**. To visualize conserved and variable regions of the TEN2 β-barrel and the LPHN3 Lec/Olf domains, we mapped the conservation of residues on the TEN2/LPHN3 complex structure, and colored residues from most conserved (magenta) to least conserved (cyan) (Fig. 3a). The interaction surfaces of TENs and LPHNs correspond to one of the most conserved regions (yellow ovals in Fig. 3a). As the TEN2 β-barrel is homologous to bacterial Tc toxins, we also analyzed the conservation between bacterial toxins by mapping the conservation of residues between bacterial toxins on the homologous bacterial TcC toxin structure (PDB ID: 4O9X)[40] and observed that the identical surface of bacterial Tc toxins is not conserved (Supplementary Fig. 5a). Our cryo-EM map thus revealed that LPHN3 binds to a highly conserved surface on the barrel of TEN2 that likely evolved to bind to LPHNs after diverging from bacterial toxins.

The Lec domain of LPHN3 belongs to the sea urchin egg lectin (SUEL) related Lec family. It adopts a kidney shape with dimensions of 20 Å × 20 Å × 50 Å, and is composed of five β-strands and a single alpha helix, interconnected by four conserved disulfide bonds[41]. In our map, the Lec domain was not as well resolved as TEN2 (Supplementary Figs. 1d, 4c). Therefore, the crystal structure of the Lec domain was docked as a rigid body without fitting the side chains. The docking of the complementary surfaces of the β-sandwich of the LPHN3 Lec domain to the concave surface of the TEN2 β-barrel creates an average interface area of 690 Å$^2$. The high affinity of the TEN2/LPHN3 complex is achieved by a combination of tentative interactions, comprised of salt bridges, hydrogen bonds, and long-range electrostatic interactions (Fig. 3b). Notably, salt bridges located at the top, middle and the bottom of the interface stabilize the interaction (Fig. 3b). The extensive network of salt bridges likely helps achieve the high affinity of the TEN2/LPHN3 complex. In addition, two residues on TEN2 (D1737 and H1738) play important roles by interacting with a disulfide bond (C36, C66) and D67 of the Lec domain (Fig. 3b). Interestingly, the cryo-EM map showed clear density of the Lec domain interacting with a glycan originating from glycosylation at N1681 (Fig. 3c). The glycan inserts into a well conserved sugar-binding pocket of the SUEL-related Lec domain of LPHN3 (Supplementary Fig. 5b), suggesting that in contrast to previous notions[41], the Lec domain of LPHN3 may still be able to bind carbohydrates.

In order to specifically abolish the interaction of TEN2 with LPHN3, and to confirm the validity of the binding interface that

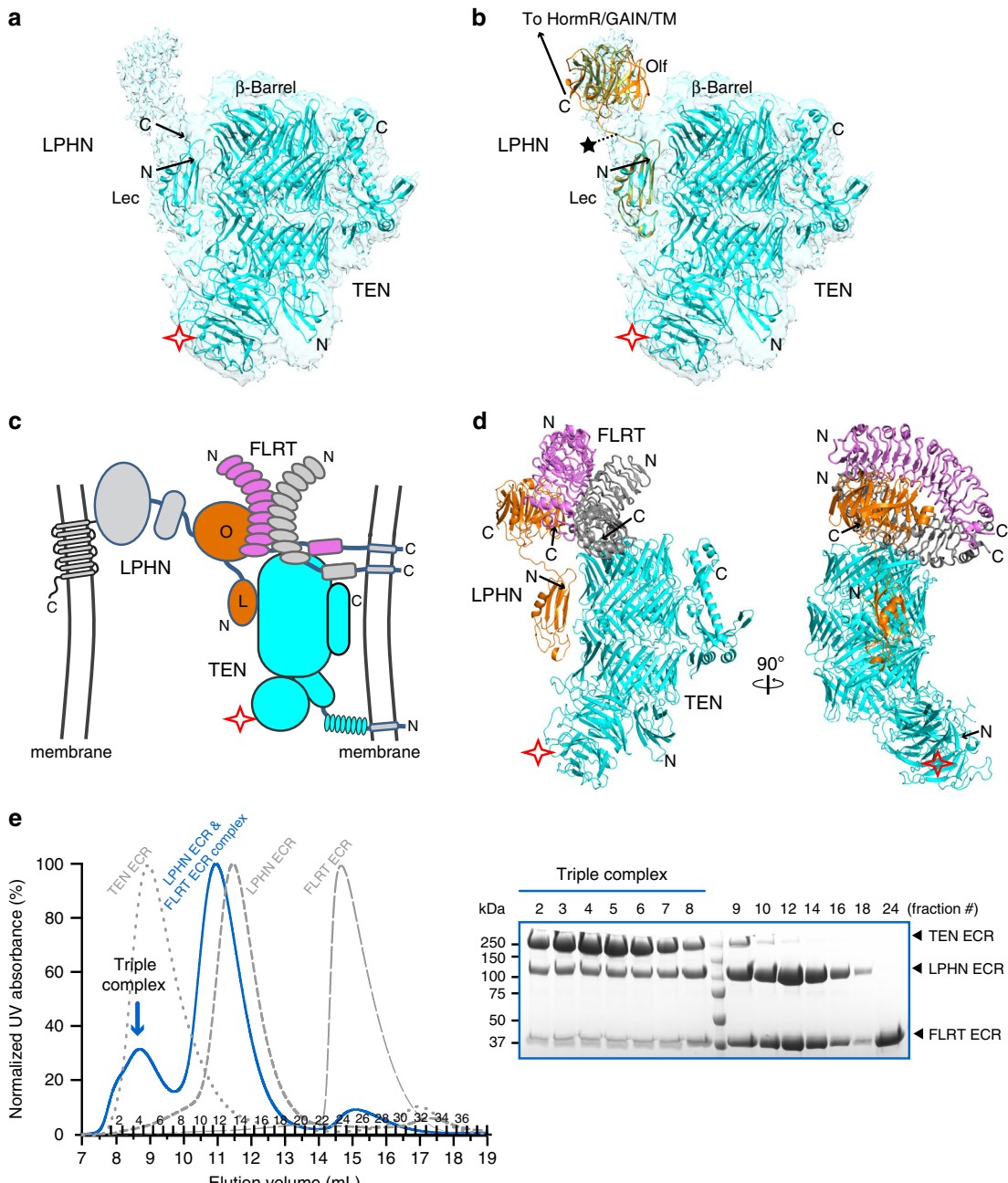

**Fig. 2 LPHN3 interacts with TEN2 and FLRT3 simultaneously. a** Continuous density C-terminal to the Lec domain of LPHN3 revealed by analysis of the cryo-EM maps at a lower threshold. **b** Manual fitting of the Olf domain from the LPHN3 Lec–Olf structure (PDB: 5AFB) into the extra EM density. Ribbon diagram of TEN2/LPHN3-Lec complex is colored cyan and the LPHN3 Lec–Olf domains are colored orange. Lec domains are superimposed. **c** Schematic diagram of the trans-cellular positioning of the TEN2/LPHN3 complex; and the formation of a ternary complex between TEN2, LPHN3, and FLRT3. **d** Superimposition of the TEN2/LPHN3-Lec–Olf complex with the LPHN3/FLRT3 complex structure (PDB: 5CMN). The Olf domains are superimposed. TEN2 and LPHN3 are colored cyan and orange, respectively. FLRT3 molecules in the FLRT3 dimer are colored magenta and gray. The second LPHN3 molecule that might be bound to the FLRT3 dimer is not shown. An UNC5 molecule that might be bound to the magenta FLRT3 is not shown. **e** SD200 size-exclusion chromatography profile of the TEN2/LPHN3/FLRT3 complex (Blue line) as compared to the profiles of individual proteins (dashed gray lines) showing that LPHN3 binds to TEN2 and FLRT3, simultaneously. Size-exclusion fractions are run on an SDS-PAGE gel. Triple complex is indicated by blue arrow. Source data are provided as a Source Data file.

we observed in the TEN2/LPHN3 complex structure, we designed mutations on full-length TEN2 that change only a few atoms on the protein surface. Several TEN2 mutations were designed, including the DHR (D1737N, H1738T, R1739T) mutation that alters residues at the LPHN3 Lec domain binding interface (Fig. 3b). To ensure that the mutant proteins are properly folded, we first examined the expression levels and surface transport of all TEN2 mutants, and exclusively used mutants that had no localization problems (Supplementary Fig. 5c). Nonpermeabilized HEK293T cells transfected with TEN2 constructs were stained with an antibody suitable to react with an extracellular tag on the proteins, and the amount of surface-exposed TEN2 was assessed

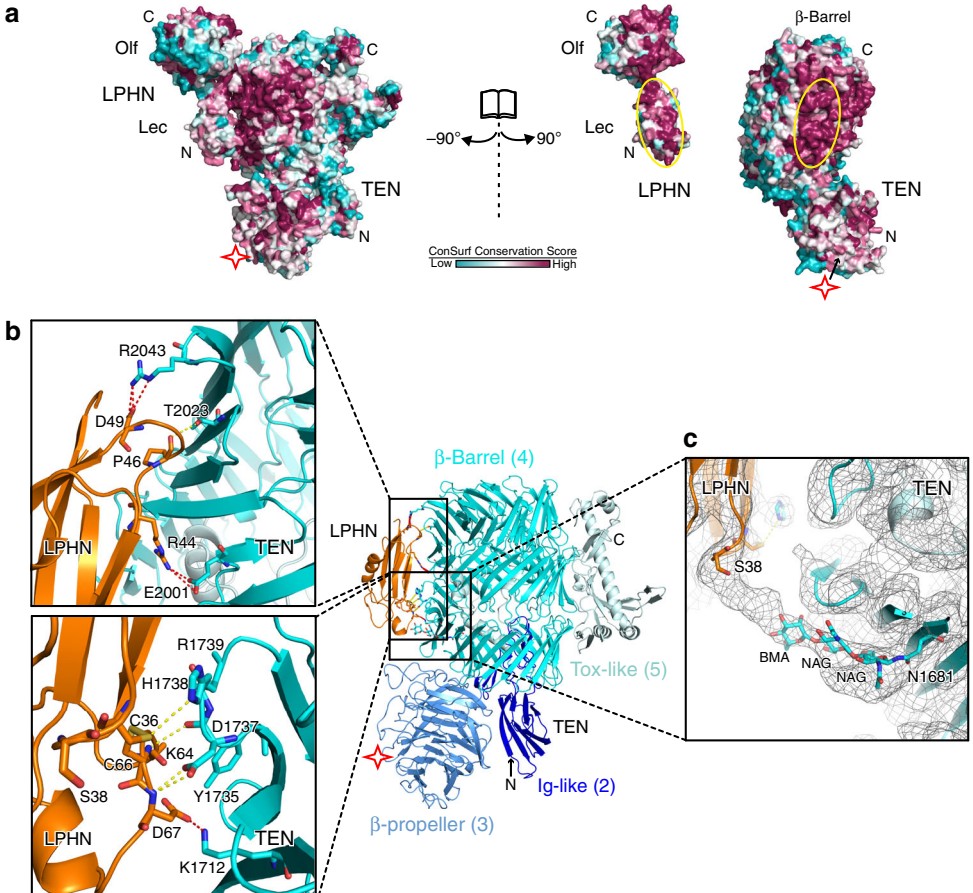

**Fig. 3 TEN2 and LPHN3 interaction is mediated by conserved residues. a** The TEN2/LPHN3 binding interface is conserved. The structure of the TEN2/LPHN3 complex is shown in surface representation on which the conservation of residues is mapped from most conserved (magenta) to least conserved (cyan) (using the ConSurf server[63]). The LPHN-binding site on TEN2 and the TEN2-binding site on LPHN3 are indicated by yellow circles. **b** Ribbon diagram of the TEN2/LPHN3 heterodimer showing the interface between TEN2 and LPHN3. Close-up view of the binding interface shows tentative residues involved hydrogen bonds and salt bridge shown by yellow dashes and red dashes, respectively. The TEN2 mutation DHR mutation (D1737N, H1738T, R1739T) which disrupt the TEN2/LPHN3 are shown as sticks. Two salt bridges between R44 and E2001; two between D49 and R2043; one between D49 and R2043; and one between D67 and K1712 are observed. **c** Close-up view for the EM density showing the close packing of the conserved N-linked glycosylation on TEN2-N1681 with LPHN3-S38 on the Lec domain. Due to the uncertainty of the glycan topology, we only modeled the first three sugars of N-glycosylation (NAG-NAG-BMA) into the density.

by indirect immunofluorescence. Importantly, the DHR mutant was properly folded and trafficked to the cell-surface. Binding experiments showed that the DHR mutant does not have the ability to interact with LPHN3. The binding experiment results are discussed in detail below within the context of geometrical restraints that act on the TEN2/LPHN3 interaction (see below).

**LPHN binding site is away from the splice site on TEN.** A seven amino acid alternatively spliced site on the β-propeller domain of TEN2 regulates the TEN2/LPHN3 interaction, and, consequently, excitatory vs. inhibitory synapse formation[2]. A striking observation from the TEN2/LPHN3 complex structure was that the LPHN3 binding site on TEN2 is located distal to the alternatively spliced sequence in the TEN2 β-propeller (Figs. 1d–f, 2c, d). This observation is very surprising because in other protein–ligand interactions regulated by alternative splicing, the alternatively spliced sequence is located at the ligand-binding interface[42,43]. Thus, in the case of the TEN2/LPHN3 interaction, alternative splicing regulates this interaction remotely in a manner that was previously not described.

We hypothesized that the membranes of two opposing cells may impose a docking geometry on TEN2 and LPHN3 that is critical for their *trans*-cellular adhesion in the extracellular space between the membranes, and that alternative splicing might control the docking geometry of TEN2. Consequently, with an altered geometry, LPHN3 may not be able to access its binding site on TEN2. This hypothesis suggests that insertion or deletion of the alternatively spliced sequence does not affect the LPHN3 binding surface on TEN2. It also suggests that when one or more membranes are removed from the experimental system, LPHN3 and TEN2 would interact with each other independent of alternative splicing because they will not be restricted by the membranes to which they are anchored. In order to test this hypothesis, we designed various experimental setups in which the restraints that act on the docking geometries of TEN2 and LPHN3 vary from high to low (Fig. 4).

First, we conducted cell-aggregation assays with HEK293 cells in which a population of HEK cells expressing full-length TEN2 are mixed with a different population of HEK cells expressing full-length LPHN3, and cell aggregation is monitored as a function of TEN2/LPHN3 interaction (Fig. 4a). These

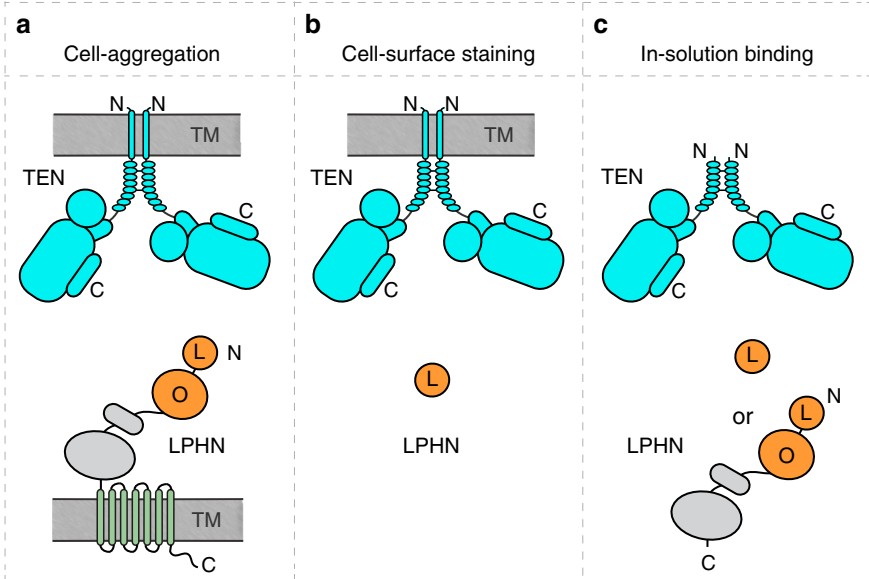

**Fig. 4 Experimental setups to investigate the effect of various restraints.** Three experimental setups with decreasing restraints on the docking geometry of LPHN3 and TEN2 during their interaction. **a** Setup for trans-cellular interaction of full-length TEN2 with full-length LPHN3 in cell-aggregation experiments. Both proteins are anchored on the cell-membranes and their mobility is restricted by their lateral diffusion within the membrane. **b** Setup for cis-like-interaction of full-length TEN2 and soluble biotinylated Lec domain of LPHN3 in either flow cytometry or cell-surface staining experiments. TEN2 is anchored on the cell-membrane, but the Lec domain of LPHN3 freely rotates in solution. **c** Setup for in-solution experiments for cis-like-interaction of soluble TEN2 and soluble LPHN3 in the absence of any membranes.

experiments mimic *trans*-cellular interaction as they detect the binding of two full-length proteins that are anchored on opposing cell membranes (referred to as *trans* hereon). Cell-aggregation experiments apply high restraints on the docking geometries of the proteins because both proteins can diffuse only laterally in two dimensions within the plane of the membrane bilayer. The cell-aggregation experiments are the best imitation for the in vivo interaction of TEN2 and LPHN3, where full-length proteins are on the cell surfaces of neighboring cells during development or synapse formation. Second, we used flow cytometry experiments and cell-surface staining experiments in which the binding of a soluble protein to a membrane-anchored protein is tested (referred to as *cis-like* hereon, although it should not be confused with commonly used meanings of *cis*) (Fig. 4b). Soluble fragments of the LPHN3 ECR were tested for their ability to bind to HEK293T cells expressing full-length TEN2 on the cell surface. These experiments apply intermediate restraints on the docking geometries of the proteins because the membrane-anchored TEN2 can diffuse only laterally in two dimensions within the plane of membrane bilayer, while the soluble LPHN3 ECR fragment can freely diffuse in three dimensions in solution (Fig. 4b). Third, we used size-exclusion chromatography in which the binding of two soluble proteins experience no restraints. Here, binding of the ECRs of TEN2 and LPHN3 is tested in the absence of any membranes (referred to as *cis-like* as well) (Fig. 4c). Importantly, any intrinsic restraints that may be originating from the intrinsic conformation of the proteins may still act on any of the above experiments. We used a combination of these experimental setups to investigate the effect of various restraints on the ability of TEN2 to interact with LPHN3.

We examined two sets of TEN2 constructs in these experimental setups: (i) TEN2 −SS carrying LPHN3 binding site mutations (TEN2 −SS DHR) was compared to WT TEN2 −SS to observe the effect of LPHN-binding site mutations on TEN2/ LPHN3 interaction (Fig. 5a). We expect that this mutant should abolish TEN2/LPHN3 interaction in all experimental setups

because it directly disrupts the LPHN-binding site on TEN. (ii) WT TEN2 +SS was compared to WT TEN2 −SS to observe the effect of inclusion of the splice insert on TEN2/LPHN3 interaction (Fig. 6a). We expect that, if the inclusion of the alternative splice insert is disrupting the binding interface on TEN2 for LPHN3, then TEN2 +SS variant should not bind LPHN3 in any of the experimental setups. However, if the insertion of the alternative splice insert is acting by a different mechanism, such as changing the docking geometry of TEN2 onto LPHN3, then, TEN2 +SS isoform may bind LPHN3 in *cis-like* setups where binding restraints on TEN2 and LPHN3 are relaxed (Fig. 4b, c).

The first set of experiments testing the effect of point mutations on the LPHN3 binding surface showed that TEN2 −SS DHR mutant was unable to bind to LPHN3 in all experimental setups, including cell-aggregation experiments (Fig. 5b), flow cytometry experiments (Fig. 5c, left), cell-surface staining experiments (Fig. 5c, right) and gel filtration experiments (Fig. 5d). These results show that this mutation destroys the binding interface on TEN2 for LPHN3 and abolishes complex formation in *trans* and *cis-like* (Fig. 5). The second set of experiments testing the effect of the alternatively spliced sequence of TEN2 on LPHN3 binding, however, displayed differential effects in *trans* and *cis-iike* experimental setups (Fig. 6). In cell-aggregation experiments, full-length TEN2 lacking the β-propeller splice insert robustly induced *trans*-cellular aggregation with full-length LPHN3 (Fig. 6b). Intriguingly, as we previously showed, inclusion of the seven amino-acid splice insert in the β-propeller eliminated trans-cellular adhesion with LPHN3 (Fig. 6b). In flow cytometry experiments, however, the affinity of TEN2 for the soluble Lec domain (that is unattached to the membranes) was not affected by inserting the seven-residue segment (Fig. 6c, left). Cell-surface staining experiments also showed that the soluble Lec domain of LPHN3 binds to both TEN2 splice isoforms, confirming the flow cytometry experiments (Fig. 6c, right). Finally, gel filtration chromatography experiments performed with the full-length

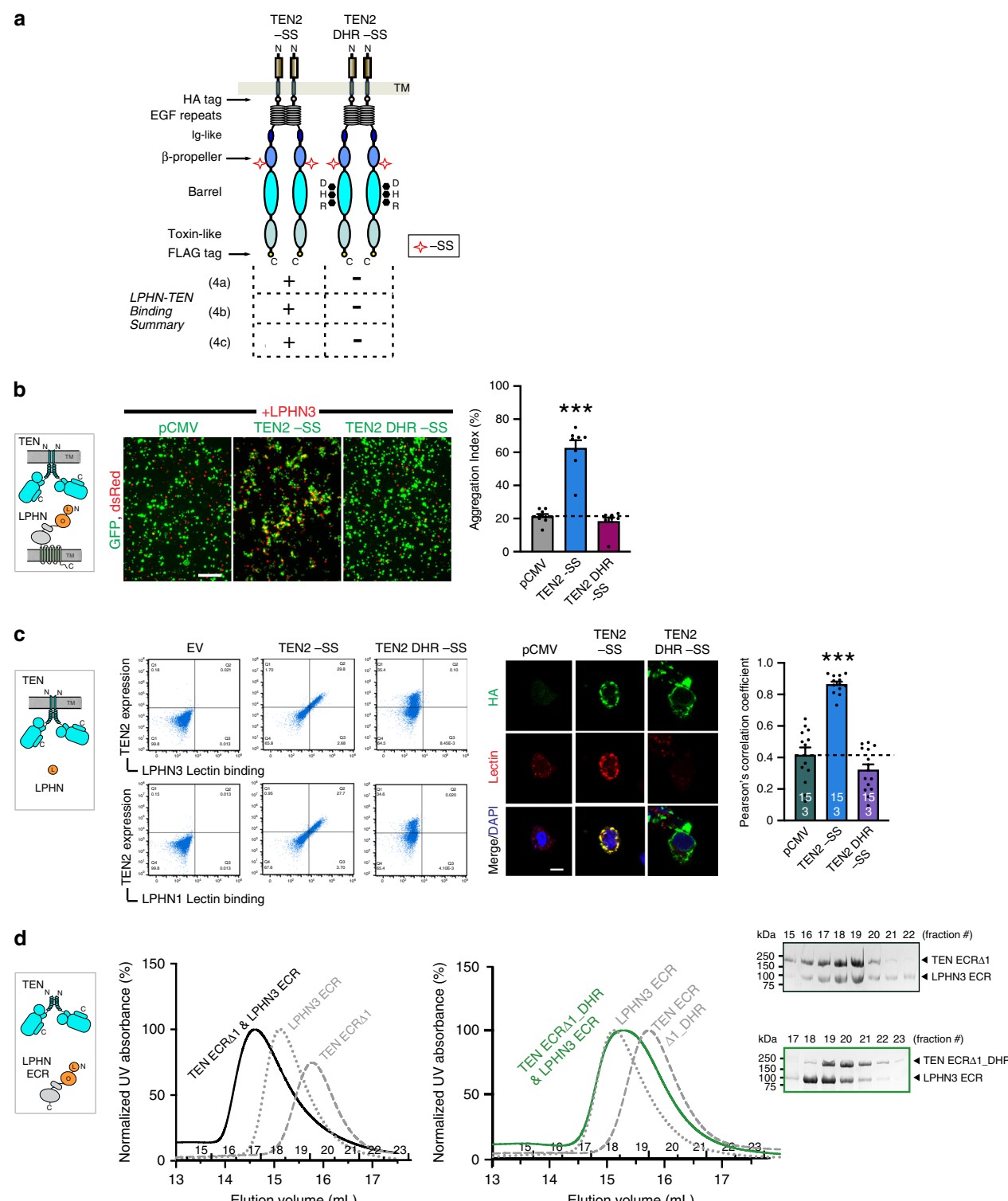

TEN2 and LPHN3 ECRs as soluble proteins unattached to membranes showed that when no restraints are applied on TEN2 and LPHN3, both TEN2 splice isoforms robustly interacted with LPHN3 (Fig. 6d). Thus, we suggest that *trans*-cellular TEN2–LPHN3 interactions are regulated by β-propeller alternative splicing likely due to conformational restrains in the context of full-length proteins.

**Binding mutants abolish excitatory synapse formation.** Previous work showed that the splice isoforms of TEN2 (TEN2 +SS and TEN2 −SS) induce different synaptic specifications in artificial synapse formation assays. In these assays, HEK293 cells expressing TEN2 variants were co-cultured with primary neurons; and inhibitory and excitatory synapse formation was monitored for pre- and postsynaptic differentiation for both types of synapses[2]. The results showed that TEN2 +SS induced GABAergic (inhibitory) postsynaptic specializations but failed to induce glutamatergic (excitatory) postsynaptic specifications[2]. On the other hand, initially, TEN2 −SS failed to recruit both excitatory and inhibitory synaptic markers[2]. However, when FLRT3,

**Fig. 5 Binding site mutations on TEN2 abolish LPHN3 binding in both trans and cis-like. a** Diagram for WT TEN2 −SS and TEN2 DHR −SS constructs. DHR mutation (D1737N, H1738T, R1739T) is on the TEN2 β-barrel located at the LPHN3-binding interface (black dots). Results for TEN2 and LPHN3 binding in different experimental setups (Fig. 4a–c) are summarized in the table. The DHR mutation breaks the interaction of TEN2 with LPHN3 in all experimental setups. **b** Representative images for cell-aggregation assays with TEN2 constructs and full-length LPHN3. WT TEN2 −SS induces cell aggregation with LPHN3, while TEN2 DHR −SS abolishes cell aggregation. HEK293 cells were co-transfected with TEN2 or LPHN3 and either tdTomato or EGFP as indicated. Scale bar indicates 100 μm. Quantification of aggregation index (%) is shown on the right (***$p < 0.001$ by one-way ANOVA). **c** TEN2 constructs expressed in mammalian cells were tested for their ability to bind soluble biotinylated LPHN3 or LPHN1 Lec domain using flow cytometry experiments (left) or cell-surface staining assays (right). The DHR mutation abolishes the cis-like interaction between TEN2 and LPHN. TEN2 construct expression was determined by HA tag fluorescence (Y axis) and purified Lec binding to TEN2-expressing cells was measured by fluorescence of DyLight attached to neutravidin (X axis). Dot plots represent the correlation between TEN2 expression and LPHN3 binding. Black cross indicates "high TEN2 expression and high LPHN3 binding" gate. Scale bar indicates 20 μm. Quantification of cell-surface-binding assays are shown next to the image (***$p < 0.001$ by one-way ANOVA). **d** Size-exclusion chromatograms showing the formation of a binary complex between soluble TEN2 −SS ECRΔ1 and full LPHN3 ECR (left, black line). Individual proteins are shown as control (dotted lines). LPHN binding mutant does not bind LPHN3 ECR (right, green line), as observed by the lack of co-elution in fractions ran on SDS-PAGE. Colors of the chromatograms and the boxes around the gels match. Data in **b** and **c** are presented as mean ± SEM, $n = 3$, and are representative of at least three independent experiments. Source data are provided as a Source Data file.

another LPHN3 ligand that on its own is also unable to induce pre- or postsynaptic specializations, was co-expressed in HEK293 cells with the TEN2 −SS, these molecules together potently induced excitatory but not inhibitory postsynaptic specializations[3] (Fig. 1c, left side).

As only the TEN2 −SS isoform is capable of interacting with LPHN3 in a *trans*-configuration, we speculated that the DHR mutation that abolishes the interaction of TEN2 with LPHN3 by demolishing the binding site should affect excitatory synapse formation, but should not impair inhibitory synapse formation that is mediated by the TEN2 +SS isoform because inhibitory synapse formation is independent of the TEN2/LPHN3 interaction. To test this hypothesis, we engineered the DHR mutation on both the full-length TEN2 −SS and the TEN2 +SS isoforms, and tested its effect on the induction of either excitatory (−SS variant) or inhibitory (+SS variant) postsynaptic specializations in the artificial synapse formation assay (Fig. 7a). As previously shown, WT TEN2 −SS induced excitatory postsynaptic specializations when co-expressed with FLRT3 (Fig. 7b, c); and WT TEN2 +SS induced inhibitory postsynaptic specializations (Fig. 7d, e)[2,3]. We observed that the DHR mutant attenuated the formation of excitatory synapses when compared with wild-type TEN2 −SS (Fig. 7b, c). However, the same mutant triggered inhibitory postsynaptic specializations similar to that of the wild-type TEN2 +SS (Fig. 7d, e), as predicted. These results indicate that LPHN3 binding mediates the excitatory synapse formation of TEN2 −SS, whereas binding of LPHN3 to TEN2 is not involved in inhibitory synapse formation.

## Discussion

Teneurins and latrophilins are multifunctional transmembrane proteins that perform important biological roles via trans-cellular interactions. The function of LPHN3 in excitatory synapse formation requires simultaneous binding of LPHN3 to both TENs and FLRTs, suggesting that a coincidence signaling mechanism mediates specificity of synaptic connections. Synaptic specificity is further regulated by alternative splicing of TEN2 because only the LPHN3-binding splice variant of TEN2 can induce excitatory synapses, but not the other variant that induces inhibitory synapses likely in a LPHN3-independent manner. A molecular understanding of the TEN2–LPHN3 complex and its critical regulation by alternative splicing to specify excitatory vs. inhibitory synapse specification is essential for progress in understanding synapse formation.

Here, we determined the cryo-EM structure of the LPHN3–TEN2 complex which revealed that the N-terminal Lec domain of LPHN3 binds to the side of the TEN2 barrel opposite

to the toxin-like domain (Fig. 1d–f). Previously, we reported that the toxin-like domain of TEN2 is needed for LPHN3 binding because a toxin domain deletion construct (TEN2 ΔTox) abolished Lec binding in cell-aggregation and cell-surface staining experiments, and lacked any defects in cell-surface localization[2]. Our further experiments suggested that this mutant is unable to bind LPHN likely because it is misfolded and escaped the protein quality control system and was still trafficked to the cell surface (Supplementary Figs. 6 and 7). During the revision of this manuscript, the structure of the chicken Ten2 in complex with mouse Latrophilin2 was published revealing a similar structure to our complex structure[44]. Both structures agree that LPHN binds to the side of the TEN2 barrel and not the toxin domain. The nearby Olf domain of LPHN3 faces away from the N-terminus of TEN2, positioning the membrane-anchored domains of LPHN3 and TEN2 opposite from each other, consistent with a trans-cellular interaction, rather than cis (Fig. 2). A FLRT3 molecule can simultaneously bind to the Olf domain of LPHN3 and form a trimeric TEN2–LPHN3–FLRT3 complex (Fig. 2). Whether the trimer may accommodate binding of a second FLRT3 molecule to enable FLRT3 dimerization or binding of an UNC5 molecule on the FLRT monomer may depend on the alternatively spliced sequence of LPHN3 between the Lec and Olf domains. FLRT3 dimerization or the FLRT3/UNC5 interaction may lead to further rearrangement of the protein-protein interaction network at the synapse.

Importantly, the LPHN3–TEN2 complex structure revealed that the LPHN3-binding site on TEN2 is away from the alternatively spliced site that is on the TEN2 propeller (Figs. 1–3), raising the question of how a seven amino acid splice insert within the >2000 amino acid ECR of TEN2 could dictate LPHN3 binding and synapse specificity without being close to the binding interface. The crystal structure of the TEN2 +SS isoform showed that the splice insert lies at the crystal contact site and likely mediate TEN homodimerization (Fig. 8a)[38]. Alternative splicing in the coding region of proteins expands the functional and regulatory capacity of metazoan genomes[45–47,48]. In addition to TEN2, numerous proteins such as DSCAMs, protocadherins, neurexins and neuroligins use alternative splicing for diversifying their functions, such as their ability to bind ligands[49–52]. In most proteins, the alternatively spliced sites localize to the ligand-binding site in order to directly enable or disturb ligand binding[42,43,53]. Thus, it is unusual that the LPHN3 binding site is localized away from the alternatively spliced sequence. In the case of TEN2, alternative splicing allows the protein to act as a switch in regulating ligand binding despite the ligand-binding site being away from the seven residue alternatively spliced site[2], and this switch disables LPHN3 binding that is required for excitatory

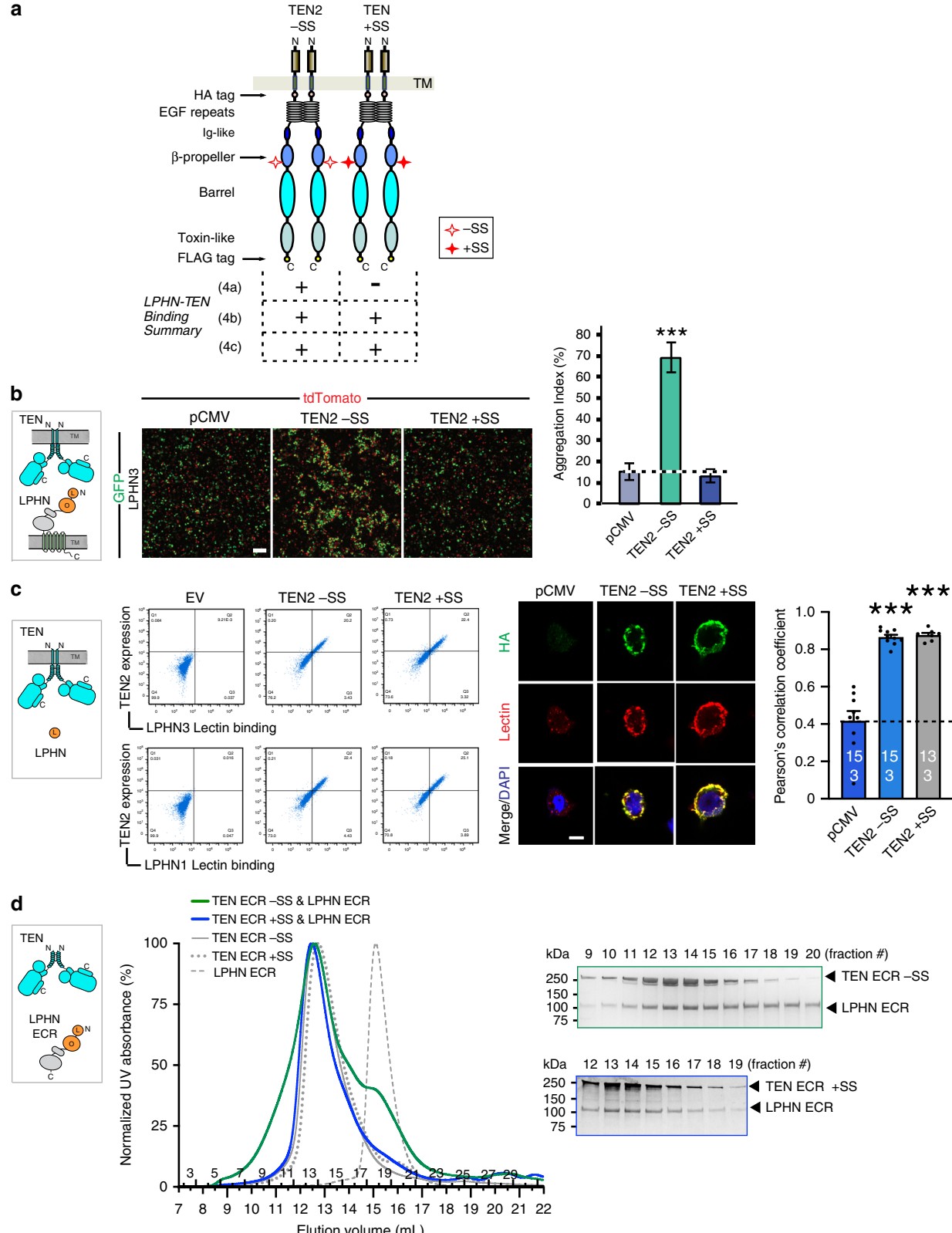

synapse formation while it likely enables (an)other interaction(s) required for inhibitory synapse formation.

Although it is intuitively difficult to understand the relationship between alternative splicing and LPHN3 binding, our synapse formation experiments demonstrated a clear requirement of TEN2–LPHN3 interaction for the excitatory synapse

specification function of TEN2 −SS, since the DHR mutation on TEN2 −SS isoform that is unable to bind to LPHN3 was unable to induce excitatory postsynaptic specializations (Fig. 7b, c). The same mutation on the TEN2 +SS isoform, however, behaved like wild-type TEN2 +SS and successfully induced inhibitory synapse formation (Fig. 7d, e). These results suggest that interaction of

**Fig. 6 Membrane anchoring restricts alternative splice-dependent interaction of TEN2 to LPHN3.** Same three experimental setups as in Fig. 4a–c were used to test the effect of alternative splicing on TEN2/LPHN3 interaction. Figure outline is identical in principle to that of in Fig. 5. **a** Diagram for WT TEN2 −SS and WT TEN2 +SS constructs that were used in the below experiments. The seven amino acid splice site on the TEN2 β-propeller is indicated by empty or filled red stars. Results for the interaction of TEN2 and LPHN3 in different experimental setups (as in Figs. 4a–c and 5b–d) are summarized in the table. The insertion of the splice site breaks the interaction of TEN2 with LPHN3 only in the cell-aggregation assays, but not in the other experimental setups. **b** Representative images for cell-aggregation assays with TEN2 −SS or TEN2 +SS and full-length LPHN3. TEN2 −SS induces cell aggregation with LPHN3, while TEN2 +SS abolishes cell aggregation. Scale bar indicates 100 μm. Figure modified from ref. [2]. **c** TEN2 −SS and TEN2 +SS expressed in mammalian cells were tested for their ability to bind soluble biotinylated LPHN3 or LPHN1 Lec domain using flow cytometry experiments (left) and using cell-surface staining assays (right). Both −SS and +SS mediate the interaction between TEN2 and LPHN in *cis-like*. Quantification of cell-surface-binding assays are shown next to the image. The cell-surface staining assays in **c** was performed in the same experiment as in Fig. 5c, and thus the control images are identical. Scale bar indicates 20 μm. (***$p < 0.001$ by one-way ANOVA.) **d** Size-exclusion chromatograms showing the formation of binary complexes between soluble full TEN2 ECR and full LPHN3 ECR (left, blue and green lines). Elution profile for individual TEN2 −SS ECR, TEN2 +SS ECR and LPHN3 ECR are shown for reference (gray lines). Both TEN2 −SS ECR and TEN2 +SS ECR bind to LPHN3 ECR (green and blue lines, respectively), as also observed by co-elution in the fractions ran on SDS-PAGE gel. Colors of the chromatograms match the colors of box around the SDS-PAGE gel. Data in **b** and **c** are presented as mean ± SEM, $n = 3$, and are representative of at least three independent experiments. Source data are provided as a Source Data file.

TEN2 with LPHN3 is required for excitatory but not for inhibitory synapse formation. Moreover, the observation that the DHR LPHN-binding mutant had no effect on the ability of TEN2 +SS to induce inhibitory postsynaptic specializations suggests a LPHN-independent mechanism that requires unidentified TEN2 interaction partners at inhibitory synapses.

Our results show that the interaction of LPHN3 with TEN2 can be disrupted in at least two ways: (1) by point mutations on the LPHN-binding interface on TEN2 (but not by mutations that are not at the interface, Supplementary Fig. 6), and (2) by insertion of the seven alternatively spliced residues in the propeller domain of TEN2. The mutagenesis of the LPHN-binding interface abolished the TEN2/LPHN3 interaction in all experimental setups as expected from a binding site mutant (Fig. 5). However, the effect of alternatively spliced site on the TEN2/LPHN3 interaction depended on whether one or both proteins experienced restraints due to their attachment to cell membranes; or they could freely rotate and tumble in solution (Fig. 6). Specifically, alternative splicing abolished the TEN2/LPHN3 interaction in cell-aggregation experiments where proteins approach each other from opposing membranes; but not in cell-surface staining or in-solution experiments where one or more proteins are in solution. Altogether these results suggest that alternative splicing regulates the TEN2/LPHN3 interaction via a mechanism that differs from disrupting the binding interface. These results enable us to suggest a model for how alternative splicing regulates TEN2 interactions and functions:

TEN2 forms a cis-dimer on the presynaptic membrane that is mediated by two disulfide bonds formed between the 2nd and 5th EGF repeats (black lines, Fig. 9) that extend the globular cytoplasmic C-terminal heads of TEN2 (TEN2 ECRΔ1) towards the opposite membrane (Figs. 8, 9). Previous cryo-EM images of the dimeric TEN2 −SS showed that the globular heads have the rotational flexibility around the EGF/head linker (arrow) that enables TEN2 −SS to sample the 3D space[2] and to successfully bind the Lec domain of LPHN3 in *cis-like* and *trans* (Fig. 9a). In this conformation, FLRT3 is also able to interact with LPHN3 and form a trimeric complex, consequently leading to excitatory synapse formation (Fig. 9a). However, the crystal structure of the TEN2 +SS isoform showed that, in the presence of the splice insert, the two globular heads form a dimer that is facilitated by the interactions between the splice inserts (Fig. 8a). The presence of the splice inserts enables the formation of two salt bridges (E1306-H1315 and E1301-R1337 in chicken Ten2, red dashed lines in Fig. 8a and black balls in Fig. 9b) and five hydrogen bonds between the β-propellers. These newly generated interactions of the propeller domains would zipper-up the molecule introducing

rigidity to the TEN2 +SS cis-dimer and restrict rotational flexibility around the EGF/head linker preventing TEN2 +SS from sampling the 3D space (Fig. 9b). On the other hand, the ECR of LPHN3 consists of two globular regions separated by a Ser-Thr-Pro rich glycosylated linker region that is reported to be semi-rigid[54]. As a result, the Lec domain of LPHN3 on the opposite membrane would have limited or no access to the LPHN-binding site on TEN2 +SS and fail to bind, although the binding site is intact and functional. In order to test the validity of this model, we introduced mutations to break the two salt bridges that are newly generated in the TEN2 +SS isoform (E1154A, H1161A, R1183A, E to A in human TEN2 splice site NKEFKHS) and to decrease the rigidity introduced by the seven amino acid splice insert (Fig. 8a). Cell-aggregation experiments showed that the TEN2 +SS salt bridge mutant restored the LPHN3 binding ability of TEN2 +SS partially (Fig. 8b) suggesting that the rigidity introduced by the salt bridges that are formed upon insertion of the splice site limits accessibility of TEN2 to LPHN3.

As alternative splicing prevents the formation of the TEN2/LPHN3 interaction by hiding the binding site rather than destroying it, it is plausible that the +SS isoform adopts a shape that enables other TEN2 interactions that the −SS isoform cannot mediate. Indeed, the trans-dimerization of TEN2 was reported to occur only by the +SS isoform[13]; and previous studies suggested that TEN2 +SS isoform should interact with unknown ligands in order to induce inhibitory synapses[2]. This structure of a teneurin–latrophilin complex in combination with our biochemical results demonstrate the clear mechanistic difference of excitatory vs. inhibitory synapse specification and lead to previously unimagined new directions in both the synapse formation and alternative splicing fields.

## Methods

**Cell culture.** High-Five insect cells (Trichoplusia ni, female, ovarian. Thermo Fisher, B85502) cultured in Insect-Xpress medium (Lonza, 04351Q) supplemented with 10 μg/mL gentamicin at 27 °C were used for production of recombinant proteins. HEK293T mammalian cells (ATCC, CRL-3216) were used for cell-surface expression assays and flow cytometry binding assays and were cultured in Dulbecco's modified Eagle's medium (DMEM; Gibco, 11965092) supplemented with 10% FBS (Sigma-Aldrich, F0926) at 37 °C in 5% $CO_2$.

**Cloning and expression in insect cells.** TEN2 splice variant Lasso (UniProt: Q9NT68-2) and LPHN (LPHN1, UniProt: O88917; LPHN3, UniProt: Q9HAR2) constructs were cloned into a pAcGP67a vector and expressed in High-Five insect cells using the baculovirus expression system. Sf9 cells (Thermo Fisher, 12659017) were co-transfected with the linearized baculovirus DNA (Expression Systems, 91-002) and the constructed plasmid using the Cellfectin II (Thermo Fisher, 10362100) transfection reagent. Baculovirus was amplified in Sf9 cells in SF900-III medium containing 10% (v/v) FBS (Sigma-Aldrich, F0926). Large-scale protein

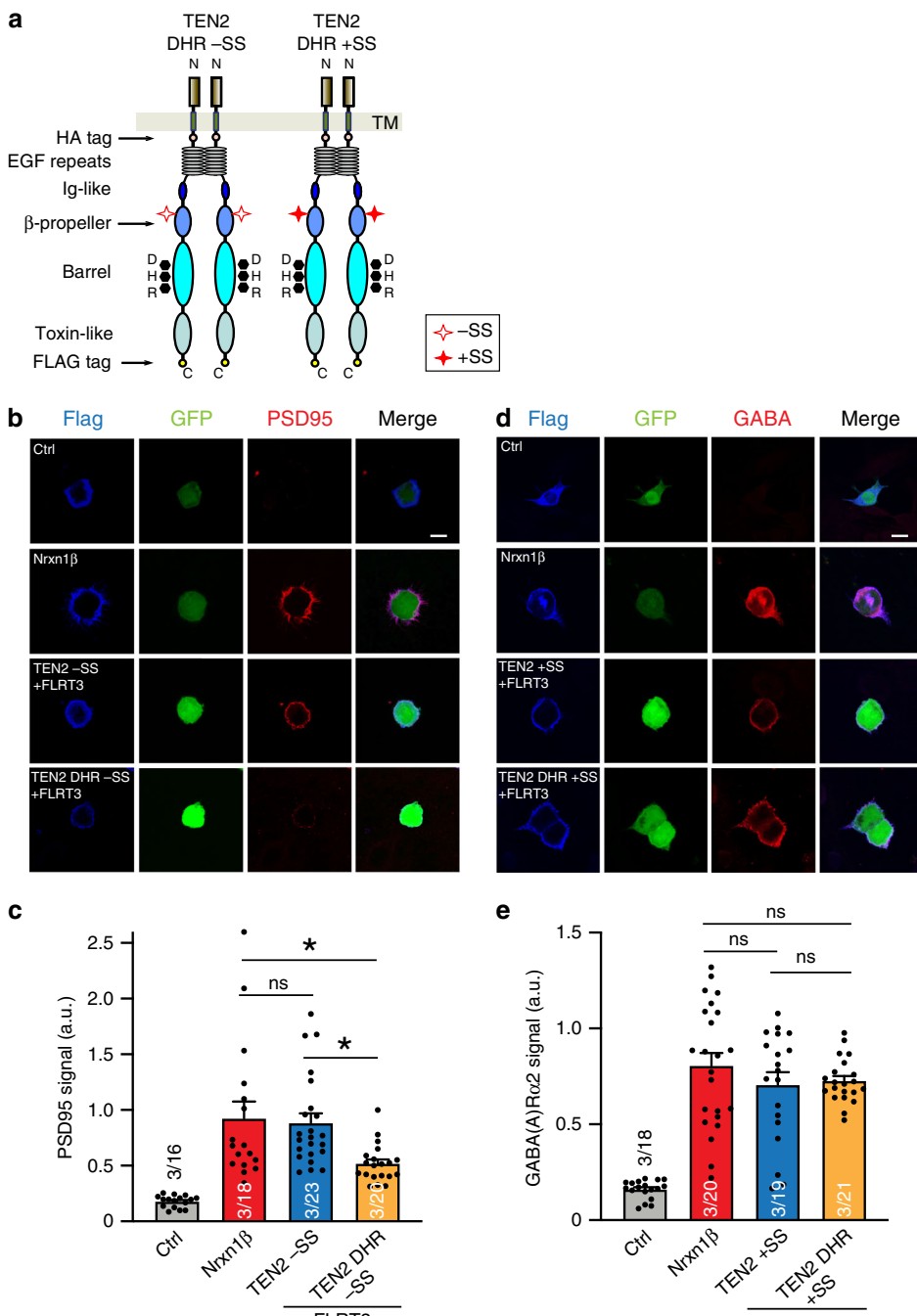

**Fig. 7 Binding site mutations on TEN2 selectively abolish excitatory but not inhibitory synapse formation. a** Diagram for TEN2 DHR −SS and TEN2 DHR +SS constructs that were used in the below experiments. The seven amino acid splice site on the TEN2 β-propeller is indicated by empty or filled red stars; DHR mutation is indicated by black dots. **b, c** Artificial synapse formation assay showing that LPHN-binding mutant (DHR) of TEN2 −SS attenuated excitatory synapse formation. HEK293T cells are co-transfected with indicated cell-adhesion molecules and GFP, and co-cultured with cortical neurons. Cultures were subsequently immunostained for the excitatory postsynaptic synapse marker PSD95. Representative images (**b**) and quantifications of PSD95 signals (**c**) are shown. **d, e** LPHN-binding mutant (DHR) of TEN2 +SS did not affect inhibitory synapse formation. Similar in **b** and **c**, except that immunostaining for the inhibitory postsynaptic synapse marker GABA(A)α2 was performed. Scale bar in **b** and **d** indicates 10 μm. Data in **c** and **e** are presented as mean ± SEM, $n = 3$, and are representative of at least three independent experiments. ns, $P > 0.05$; *$P < 0.05$ (one-way ANOVA). Source data are provided as a Source Data file.

expression was performed by infection of High Five cells (Thermo Fisher, B85502) in Insect-XPRESS medium (Lonza, 12-730Q) medium at a cell density of $2.0 \times 10^6$ cells/ml for 72 h at 27 °C.

For the structural studies, TEN2 ECRΔ1 (residues T727-R2648) and LPNH3 ECR (residues S21-V866) were cloned with carboxyl-terminal 6XHis-tags separately and co-expressed in High-Five insect cells. The following primers were used for amplification of High Five cells expressed human TEN2 ECRΔ1: F: 5′-

CATTCTGCCTTTGCGGCGGATCCCACTTCCTGTGCTGATAACAAGGAT AATGAG-3′ and R: 5′-GGATCAGATCTGCAGCTTAGTGATGGTGATGGTGA TGCCTCTTTCCCATCTCATTCTGTC-3′. The following primers were used for amplification of High Five cells expressed human LPHN3 ECR: F: 5′-CATTCTG CCTTTGCGGCGGATCCCTCCCGCGCACCCATTCC-3′ and R: 5′-GGATCAGA TCTGCAGCTTAGTGATGGTGATGGTGATGCACGTCCAGCAGCAGATCG TG-3′. Seventy-two hours after viral infection, the medium containing secreted

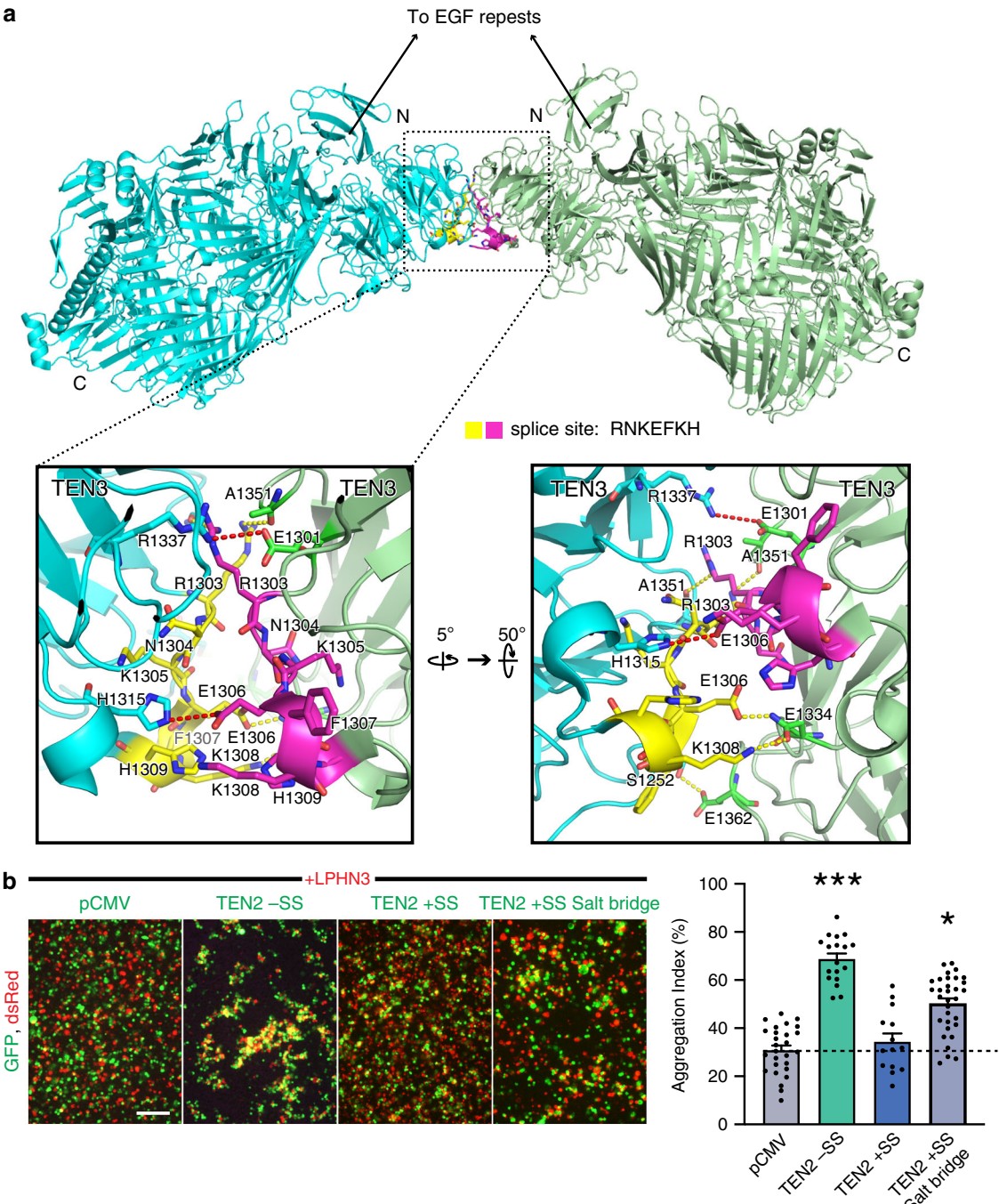

**Fig. 8 Alternatively spliced insert within the β-propeller mediates the TEN2 +SS dimer interface. a** Structure of TEN2 +SS dimer shows the splice inserts from each protomer (yellow and magenta residues) creates a binding interface and leads to TEN2 dimerization via the β-propeller. Close-up views of the dimer interface show two salt bridges and five hydrogen bonds are at the interface. One of the salt bridges is directly mediated by the glutamate (E1306) within the splice insert NKEFKHS and four of the hydrogen bonds also require splice site residues. Two salt bridges align almost parallel to each other and to the disulfide bonds between the EGF repeats and restricts the conformational flexibility of the TEN2 head significantly (see Fig. 9). The N-termini of both protomers face the same direction towards the EGF repeats, and thus, the dimer is positioned as a cis-dimer that will extend the zippering of the already existing EGF-mediated cis-dimer, although it was reported to form as a trans-homodimer, previously[38]. TEN protomers (PDB: 6FB3) are colored as cyan and palegreen, respectively, and splice sites are colored as yellow and magenta, respectively. **b** Cell-aggregation experiments show the LPHN3 binding ability of TEN2 +SS is partially restored when the two salt bridges are broken in the TEN2 +SS mutant (*$P \leq 0.05$; ***$P \leq 0.001$; by one-way ANOVA). Data are presented as mean ± SEM, $n = 3$, and are representative of at least three independent experiments. Source data are provided as a Source Data file.

glycosylated proteins was collected and centrifuged at 900 g for 15 min at room temperature. The supernatant was transferred into a beaker and mixed with (final concentrations): 50 mM Tris pH 8.0, 5 mM CaCl$_2$ and 1 mM NiCl$_2$ and stirred for 30 min. After centrifugation at 8000 g for 30 min, the clarified supernatant was incubated with nickel-nitrilotriacetic agarose resin (QIAGEN) for 3 h at room temperature. The resin was collected with a glass Buchner funnel and rinsed with HBS buffer containing 20 mM imidazole, then transferred to a poly-prep chromatography column (Bio-rad). The protein was eluted with HBS buffer containing 200 mM imidazole and run on size-exclusion chromatography (Superdex 200 10/300 GL; Superose 6 Increase 10/300 columns; GE Healthcare),

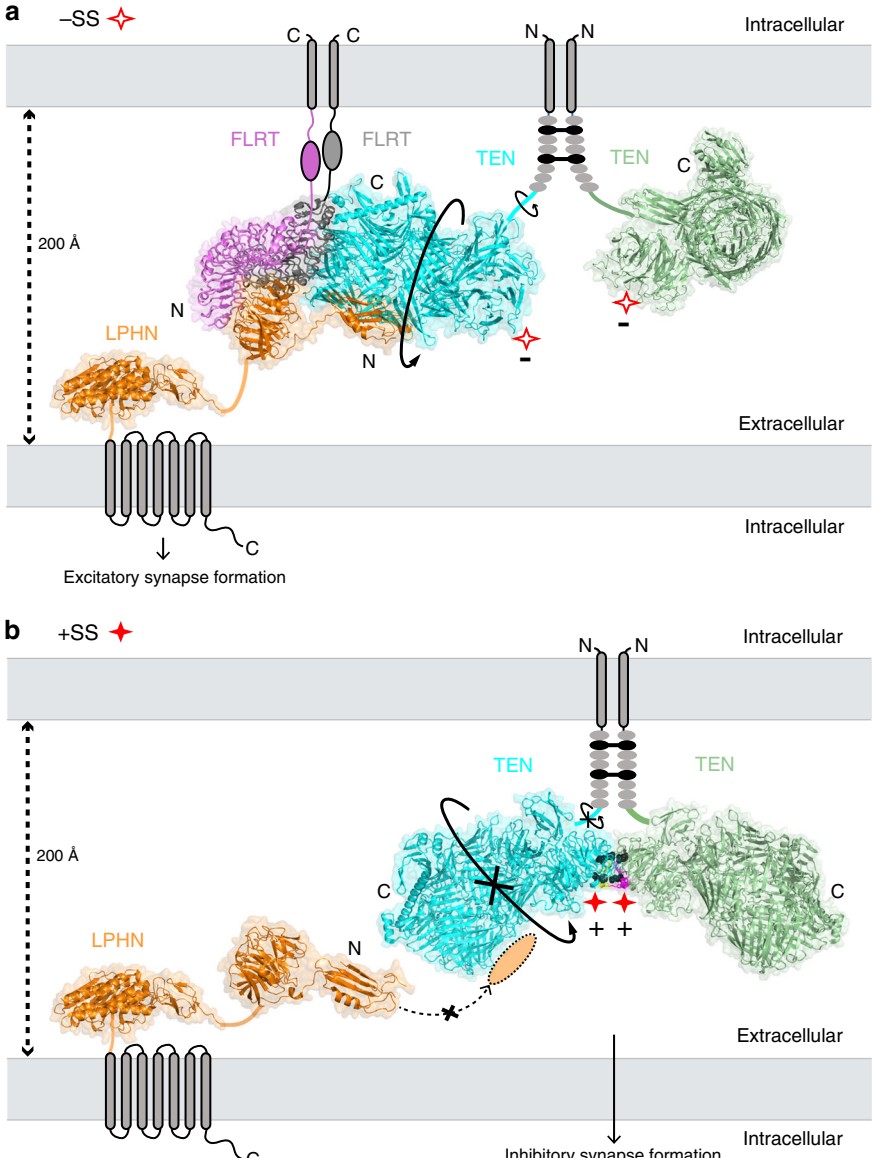

**Fig. 9 Model for the splice variant-dependent interaction of TEN2 with LPHN3.** The model depicts how alternative splicing acts as a molecular switch to determine which adhesion partner TEN2 binds to and, accordingly, which type of synapse TEN2 specifies. Both TEN2 isoforms form a cis-dimer on the presynaptic membrane through two disulfide bonds formed between the 2nd and 5th EGF repeats (black sticks). **a** TEN2 −SS isoform has rotational flexibility (arrows) that enables TEN2 to find the correct docking geometry in order to bind to the Lec domain of LPHN3 expressed on the neighbor cell[2]. Such rotational flexibility also allows FLRT3 to bind to the Olf domain of LPHN3 and, altogether, to induce excitatory synapse formation. The DHR mutation breaks the interaction of TEN2 −SS with LPHN3 and abolishes excitatory synapse formation. **b** The TEN2 +SS isoform does not have rotational flexibility around the linker between the EGF repeats and the rest of the extracellular head as observed in the crystal structure of the TEN2 +SS isoform, which shows that the splice insert mediates a dimeric interaction between the two TEN2 +SS protomers[38] (Fig. 8a, PDB ID: 6FB3). Instead, the TEN2 +SS protomers are zipped-up due to the additional two salt bridges (black balls) between the propellers of the TEN2 cis-dimer. Thus, the LPHN-binding site on TEN2 +SS is not at the right docking geometry to interact with the Lec domain of LPHN3 expressed on the neighboring cell (though it can still bind soluble Lec domain). The geometry of TEN2 +SS likely enables other hetero- or homophilic protein interactions that were not possible in the −SS isoform, such as TEN2 trans-homodimerization, and mediates inhibitory synapse formation. The DHR mutation on TEN2 +SS has no effect on these unknown interactions and thus, does not affect the ability of TEN2 +SS to induce inhibitory synapses. Model partially drawn to scale. The LPHN ECR structure (orange), is based on Lec and Olf domain structure (PDB: 5AFB), connected by a STP-rich stalk to the GAIN and HormR domain structure (PDB: 4DLQ). TEN2 protomers are colored as cyan and palegreen (PDB: 6CMX and 6FB3), FLRT protomers are colored as magenta and gray (PDB: 5CMN).

purified in a final buffer comprised of 10 mM Tris pH 8.5, 150 mM NaCl. For the flow cytometry binding assays, LPHN1 Lec (residues S26-Y131) and LPHN3 Lec (residues S21-Y126) were cloned with carboxyl-terminal 6XHis-AVI-tags and captured on nickel-nitrilotriacetic resin as described above. Following a wash with HBS buffer containing 20 mM imidazole, final concentrations of 50 mM Bicine pH 8.3, 100 mM NaCl, 10 mM Mg-acetate, 10 mM ATP, 0.5 mM biotin and 5 mM BirA were added to the resin, which was then rotated for 2 h at 27 °C. After removing residual BirA and ATP by washing with HBS buffer containing 20 mM

imidazole, the biotinylated lectin was eluted with HBS buffer containing 200 mM imidazole. Purified protein was applied to size-exclusion chromatography. The efficiency of biotinylation was assessed using a streptavidin bead pulldown assay.

**Cloning and expression in mammalian cells.** Full-length TEN2 (residues M1-R2648) construct and TEN2 mutants (DHR mutant: D1737N, H1738T, R1739T; LR mutant: L1990N, R1992T) bearing HA-tag (inserted between K405/E406) and

carboxyl-terminal FLAG-tag were cloned into a pcDNA3.1 vector for cell-surface expression assays and flow cytometry binding assays in HEK293T cells. The following primers were used for amplification of HEK cells expressed human TEN2 ECR: F: 5′-GGATGACGACGATAAAGGCGGTAAGCTTAGCCCACCTCTC-3′ and R: 5′-TTACTTATCGTCGTCATCCTTGTAATCCCTCTTTCCCATCTCATT CTGTCCTT-3′. The following primers were used for amplification of HEK cells expressed human TEN2 ΔTox: F: 5′-GGATGACGACGATAAAGGCGGTAAGC TTAGCCCACCTCTC-3′ and R: 5′-TTACTTATCGTCGTCATCCTTGTAATCTT CATAGGGAGGAGGCACGAAATACAT-3′. The following primers were used for amplification of HEK cells expressed human TEN2 ΔToxΔBarrel: F: 5′-GGATGA CGACGATAAAGGCGGTAAGCTTAGCCCACCTCTC-3′ and R: 5′- TTACTTA TCGTCGTCATCCTTGTAATCGAAGGCATTAAGAACAGGCTTGTTC-3′. TEN2 DHR mutants were generated using a standard two-step PCR-based strategy with primers: F: 5′-ATTCGGACTGAAAAGATCTATGATAACACCACGAAGTT CACCCTGAGGATCATTTATG-3′ and R: 5′-CATAAATGATCCTCAGGGTGA ACTTCGTGGTGTTATCATAGATCTTTTCAGTCCGAAT-3′. TEN2 LR mutants were generated using a standard two-step PCR-based strategy with primers: F: 5′- AGTGAGACTCCCCTCCCCGTTGACAACTACACCTATGATGAGATTTCT GGCAAGGTG-3′ and R: 5′-CACCTTGCCAGAAATCTCATCATAGGTGTAG TTGTCAACGGGGAGGGGAGTCTCACT-3′.

**Flow cytometry**. HEK293T cells were cultured in 6-well plates and were transfected 2 μg cDNA using LipoD293 transfection reagent. Cells at 50-60% confluence were transiently transfected as follows: 2 μg cDNA was diluted in 50 μl serum-free DMEM, and 3 μl LipoD293 transfection reagent (SignaGen, SL100668) was diluted with 47 μl serum-free DMEM. The diluted LipoD293 was added to the diluted cDNA and incubated for 10 min. Then, the transfection mixture was added dropwise to each well. The cells were detached using citric saline solution (50 mM sodium citrate, 135 mM KCl) after 48 h incubation and washed with PBS + 2% BSA. To test TEN2 WT and mutant cell-surface expression, cells were stained with a primary antibody mixture: mouse anti-FLAG M2 (Sigma, F3165) 1:1000 and rabbit anti-HA (Life Technologies, 715500) 1:1000 for 30 min at room temperature. After wash with PBS + 2% BSA, cells were stained with a secondary antibody mixture: donkey anti-mouse Alexa Fluor 488 (nvitrogen, A21202) 1:3000 and goat anti-rabbit Alexa Fluor 647 (Invitrogen, A32733) 1:3000 for 30 min. After washing, cell pellets were resuspended in PBS + 2% BSA immediately before flow cytometry data acquisition (Accuri C6 flow cytometer, 10000 events measured) after washing. Acquired data were analyzed using the FlowJo analysis software (FlowJo LLC).

For the binding assays, His-Avi-tagged Lec was captured on nickel-nitrilotriacetic acid resin and purified as described above. Biotinylated Lec was tetramerized and fluorescently labeled through incubation with NeutrAvidin DyLight 488 (Thermo, 22832) on ice for 20 min. Cultured cells expressing HA-tagged TEN2 were detached and then washed as described above. Next, the cells were stained with rabbit anti-HA 1:1000 antibody and, following two wash cycles, stained with goat anti-rabbit Alexa Fluor 647 antibody in the presence of the 100 nM NAV488 labeled Lec mixture. The following primers were used for amplification of His-Avi-tagged human LPHN1 Lec: F: 5′-CGGCGGCGCATTC TGCCTTTGCGGCGAGCCGGGCTGGACTCCCATTTGG-3′ and R: 5′-CTTCTG AGCCTCGAAAATATCATTAAGACCGCGGTAAGGGACACAGTCGTACT GC-3′. The following primers were used for amplification of His-Avi-tagged human LPHN3 Lec: F: 5′-GGCGGCGCATTCTGCCTTTGCGGCGTCCCG CGCACCCATTCCTATGGCCG-3′ and R: 5′-TTCTGAGCCTCGAAAATATCA TTAAGACCGCGATATGGCACGCACTCGTACTGCACT-3′.

**Cell-aggregation assays**. HEK293T cells (ATCC) were grown to 90% confluence in a T-75 flask. Cells were trypsinized with 3 mL 0.05% trypsin-EDTA (Gibco, 25300-054) and resuspended to 10 mL with DMEM/10% FBS/1% Penicillin–Streptomycin media (Complete DMEM). Three-hundred μL of the cell suspension was added to each well of a 6-well plate containing 3 mL of Complete DMEM media and incubated overnight at 37 °C. Cells in each well were then co-transfected with 2 μg of either pCMV (empty vector) + pEmerald, pCMV LPHN3 + pEmerald, pCMV (empty vector) + pCMV dsRed, or dsRed and the indicated TEN2 construct using the Calcium Phosphate method. All cDNAs were encoded in the pCMV5 or pcDNA3 vector and driven by the CMV promoter. Three days after transfection, the media was aspirated and cells were gently washed with 2 mL of PBS. Cells were resuspended by adding 1 mL of Resuspension Solution (PBS containing 1 mM EGTA) and then incubated for 5 min at 37 °C. Fifteen μL of 1 mg/20 μL DNASe (Sigma, D5025) was then added to each well and cells were triturated by pipetting up-and-down (16 times) in each well to resuspend cells off the plate bottom and create single-cell suspensions. Cells were then transferred to a new Eppendorf tube and another 15 μL of DNAse solution was added to each sample. Cells were mixed in 1:1 ratio by adding 70 μL of pCMV (empty vector) + pEmerald or LPHN3 + pEmerald with 70 μL of pCMV (empty vector) + pCMV-dsRed or dsRed and TEN2 Construct + dsRed in a new Eppendorf that contained 360 μL of Incubation Solution (DMEM containing 50 mM HEPES-NaOH pH 7.4, 10% FBS, 10 mM CaCl₂ and 10 mM MgCl₂) for a final volume of 500 μL. The mixture was triturated and the entire volume was transferred to one well in a non-coated 12-well plate (Costar, 3737). Images were taken immediately (time = 0) using a Leica Fluorescent DMIL LED Microscope with a 10x objective. Cells were then placed on a shaking incubator at 125 rpm at 37 °C for 20 min and imaged again (time = 20).

Aggregation index at time = 20 was calculated using ImageJ, measuring the percentage of signal/frame occupied by cells forming complexes of two or more cells relative to the total signal of the frame.

**Cell-surface-binding assays**. HEK293T cells (ATCC) were grown to 90% confluence in a T-75 flask. Cells were trypsinized with 3 mL 0.05% trypsin-EDTA (Gibco, 25300-054) and resuspended to 10 mL of DMEM + 10% FBS + 1% Penicillin–Streptomycin (complete DMEM) media. Fifty μL of cell suspension was added to each well of a 24-well plate that contained a Matrigel-coated coverslip and 1 mL complete DMEM and incubated overnight. Cells were then co-transfected with 1 μg of either empty pCMV, wild-type Teneurin 2 or the indicated mutant Teneurin construct and 1 μg of pEmerald using the Calcium Phosphate method and incubated for 2 days at 37 °C. Transfection media was gently removed and 500 μL of chilled DMEM containing 250 μM of purified, Avi-fusion, biotinylated, rat LPHN1 Lec or human LPHN3 Lec was added to each well. Plates were wrapped in foil and incubated overnight at 4 °C to reduce endocytosis, with gentle shaking. This was performed essentially as described in [55]. Cells were gently washed 2x using 1 mL of PBS and fixed with 300 μL of ice-cold 4% PFA/4%sucrose/PBS. Plates were wrapped in foil and incubated for 20 min at 4 °C during the fixation. Cells were gently washed 3× using 1 mL of room temperature PBS and blocked with 300 μL of 5% BSA (Sigma, 10735086001)/PBS (blocking buffer) for 1 h at room temperature. Bound biotinylated Lec was detected by immunofluorescence using 300 μL per well of Streptavidin (AlexaFluor-555 conjugated, Invitrogen, S21381, at 1:10,000 dilution) diluted into blocking buffer for 1 h at room temperature. Cells were gently washed 3× with 1 mL of PBS. Cells were re-blocked, and HA-tagged, surface Teneurins were detected by adding 300 μL of rabbit anti-HA antibody (Cell Signaling Technologies, 3724) at 1:1,000 dilution in blocking buffer. Cells were gently washed 3× with 1 mL PBS. Goat anti-rabbit secondary antibodies (Alexa-Fluor 633 conjugated, Invitrogen) and DAPI (Sigma, 10236276001) staining was done for 30 min at 1:10,000 and 1:5,000, respectively, in blocking buffer, followed by 3× gentle washes with 1 mL of PBS. Coverslips were mounted onto slides (UltraClear microscope slides Denville Scientific, M1021) in mounting media (Fluoromount-G, Southern Biotech, 010020). Images were acquired using a Nikon A1 Eclipse Ti2 confocal microscope with a ×60 oil-immersion objective, operated by NIS-Elements AR acquisition softw×are. The same confocal acquisition settings were applied to all samples of the experiment. Collected z-stacks at a 0.4 μm z-step size were analyzed blindly using Nikon Elements Analysis software. Co-localization was calculated using the Pearson's correlation coefficient of Lec-Streptavidin-555 to Teneurin-HA-633 emission.

**Artificial synapse formation assay**. HEK293T cells were transfected with the expression vectors of the cell-adhesion molecules. 24 h later, HEK293T cells were co-cultured with cultured cortical neurons (DIV16) from P0 mice. After 24 h, cells were fixed with 4% PFA and immunostained with rabbit anti-Flag (Sigma; 1:1000 both) together with mouse anti-PSD95 (Sysy, 124011, 1:500) or mouse anti-GABAA α2 (Sysy; 224211, 1:500) respectively. Images were collected with a Nikon A1 confocal microscope using a ×60 objective. A human NPR mutant (1-118 aa of the full-length protein) which comprises an A domain containing low-complexity sequences is used as the negative control in the artificial synapse formation assays. The signals of the synaptic markers that were recruited to the surface of the HEK293T cells were quantified using Image J. Normalized values equal the fluorescent intensity of the synaptic marker that was examined (GABAα2/PSD95) / the fluorescent intensity of the Flag-tagged protein expressed in the HEK cells (TENs /Nrn1β).

**Cryo-EM data acquisition**. 2.5 μl purified human TEN2 ECRΔ1 and human LPHN3 ECR complex (0.22 mg/ml) was applied on glow-discharged holey carbon grids (Quantifoil R1.2/1.3, 300 mesh), and vitrified using a Vitrobot Mark IV (FEI Company). The specimen was visualized using a Titan Krios electron microscope (FEI) operating at 300 kV and equipped with a K3 direct electron detector (Gatan, Inc.). Images were recorded with a nominal magnification of ×81,000 in super-resolution counting mode, corresponding to a pixel size of 0.54 Å on the specimen level. To maximize data collection speed while keeping image aberrations minimal, image shift was used as imaging strategy using one data position per hole with four holes targeted in one template with one focus positio. In total, 4967 images with defocus values in the range of −1.0 to −2.5 μm were recorded using a dose rate of 14.6 electrons/Å²/s. The total exposure time was set to 4.2 s with frames recorded every 0.105 s, resulting in an accumulated dose of about 60.1 electrons per Å² and a total of 40 frames per movie stack.

**Image processing and 3D reconstructions**. Stack images were subjected to beam-induced motion correction using MotionCor2[56]. CTF parameters for each micrograph were determined by CTFFIND4[57]. Particle selection, two- and three-dimensional classifications were initially performed on a binned dataset with a pixel size of 4.32 Å using RELION-3[58]. In total, 4,475,958 particle projections were selected using automated particle picking and subjected to reference-free two-dimensional classification to discard false-positive particles or particles categorized in poorly defined classes, resulting in 3,307,148 particle projections for further processing. The initial 3D maximum-likelihood-based classification was performed

on a binned dataset with a pixel size of 4.32 Å using the previously reported TEN2 structure[2] as the reference model. The detailed data processing flow is shown in Supplementary Figs. 2 and 3. Briefly, for Tenurin–Letrophilin complex, 1,309,684 particles that showed well-defined density of Lec domain were selected after initial rounds of 3D classification. Then, two rounds of focused 3D classification with mask around Lec domain were performed without alignment. 3D refinement and post-processing was performed on the best class with clear features for the Lec domain (Supplementary Fig. 2). The final map for TEN2_Lec was resolved at 2.97 Å (Supplementary Fig. 2). To resolve domain 3, the same dataset was reprocessed with a total of 1,137,765 particles showing well-resolved domain 3. Two rounds of focused 3D classification with mask around domain 3 were performed, followed by 3D refinement and post-processing. The final map for TEN2_domain 3 was resolved at 3.07 Å (Supplementary Fig. 3).

Reported resolutions are based on the gold-standard Fourier shell correlation (FSC) using the 0.143 criterion (Supplementary Fig. 1c). All density maps were corrected for the modulation transfer function (MTF) of the K3 direct detector and then sharpened by applying a temperature factor that was estimated using post-processing in RELION-3. Local resolution was determined using ResMap[59] with half-reconstructions as input maps (Supplementary Fig. 1d).

**Model building and refinement**. Model building was based on the structure of human TEN2 ECR (PDB: 6CMX) and the Lec domain from human LPHN3 (PDB: 5AFB and 5FTT). The models were first docked into the EM density maps using Chimera[60] and then manually checked and adjusted residue-by-residue to fit the density using COOT[61]. The ECR of TEN2 was built based on that of chicken TEN2 (PDB: 6FB3) and manually adjusted to human sequence and splice form. Note that the Lec domain was not as well resolved as TEN2, so it was docked as a rigid body without fitting and manipulating the side chains. Both maps (TEN2/LPHN3 and TEN2 focusing on domain 3) were used for model building. There is a slight shift between the two maps from reconstruction, so they were aligned based on TEN-Lec before model building. The final model containing both ECR of TEN2 and Lec domain of LPHN3 was subjected to global refinement and minimization in real space using the phenix_real_space_refine module in Phenix[62] first against TEN2 domain 3 map while keeping the Lec domain as a rigid body, and then against TEN2-Lec map while keeping the domain 3 as a rigid body. FSC curves were calculated between the resulting model and either maps using Phenix M-triage (Supplementary Fig. 1). The final model statistics are provided in Table 1.

Nine N-linked glycosylation sites (on residues N1490, N1586, N1647, N1681, N1766, N1867, N2071, N2211, N2522.) and five disulfide bonds (C1394-C1402), (C1396-C1404), (C1106-C1109), (C1210-C1218), (C1277-C1330) were observed in TEN2.

**Quantification and statistical analysis**. Error bars in Figs. 5, 6, 7, 8 and Supplementary Fig. 6 represent means ± SEM. Each measurement was repeated at least three times independently. Data were analyzed using software GraphPad Prism and ImageJ.

**Reporting summary**. Further information on research design is available in the Nature Research Reporting Summary linked to this article.

## Data availability

The cryo-EM density map has been deposited in the Electron Microscopy Data Bank (https://www.ebi.ac.uk/pdbe/emdb/) under accession code EMD-21205 and the model coordinates have been deposited in the Protein Data Bank (http://www.rcsb.org) under accession number PDB 6VHH. Data supporting the findings of this manuscript are available from the corresponding authors upon reasonable request. A reporting summary for this Article is available as a Supplementary Information file. The source data underlying Figs. 2e, 5b–d, 6c, d, 7c, e, 8b and Supplementary Figs. 6b, c, and 7 are provided as a Source Data file.

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

## Acknowledgements

The authors thank Tobin R. Sosnick for generously providing DesG tag construct to improve behavior of the protein, Anthony Kossiakoff, Engin Ozkan for the use of flow cytometer, Georgios Skiniotis and Moran Shalev-Binami for initial data collection. This research was in part, supported by the National Cancer Institute's National Cryo-EM Facility at the Frederick National Laboratory for Cancer Research under contract HSSN261200800001E. This work was supported by grants R01 GM120322 (to D.A.) and R01 GM134035-01 (to D.A.), Chicago Biomedical Consortium Catalyst Award C-086 (to M.Z.), and the American Heart Association grant #19POST34380439/Jingxian Li/2019-2020 (to J.X.).

## Author contributions

J.L. and D.A. designed all experiments and interpreted results. J.L. cloned, expressed and purified TEN2/LPHN3 complex, carried out TEN2 related biochemical characterizations and specimen screening, designed and performed the HEK cell expression and flow cytometry binding assays. Y.X. and M.Z. performed cryo-EM data collection and map calculation, model building and refinement. Y.X. and M.Z. carried out structural analysis with assistance from J.L. and D.A. T.C.S., R.S., X.J., and S.C designed and performed the cell-aggregation assays, cell-surface staining assays and synapse formation assays. S.P.K. participated in TEN2 alternative splice site related biochemical characterizations. M.P. and K.L. assisted in specimen screening by EM. D.A. wrote the paper with assistance from M.Z. T.C.S. J.L., and Y.X. D.A. and M.Z. supervised the project.

## Competing interests

The authors declare no competing interests.
