## [Peer Review File · Nature Communications]

Reviewers' Comments:

Reviewer #1:

Remarks to the Author:

The present manuscript follows a publication by the same authors last year (Li et al, Cell, 2018). In that publication, the authors reported the structure of the TEN2 extracellular region and showed that alternative splicing within this region regulates trans-cellular adhesion to LPHN3, and consequently synapse specification. Using deletion mutants, the LPHN binding site on TEN2 was mapped to the toxin-like domain, >80 Å away from the synapse-specifying splice insert. It was suggested that "alternative splicing may facilitate a conformational change of the entire [extracellular region] that renders the LPHN binding site inaccessible to the full-length LPHN on the neighboring cell".

In the present manuscript, the authors have determined the structure of a soluble TEN2-LPHN3 complex, which shows that the LPHN3 lectin domain does not bind to the TEN2 toxin-like domain as previously reported, but to the side of the β -barrel. The new (and likely correct) binding site is closer to the splice site, but still too distant for a direct effect on LPHN binding. Using a series of elegant interaction experiments, the authors show that the splice insert in TEN2 has no effect on LPHN binding in solution, but that it abrogates binding in a trans-cellular aggregation assay. A triple mutation at the new LPHN binding site selectively abolishes excitatory (but not inhibitory) synapse formation. The findings are combined into a new model, in which the splice insert promotes the formation of a tight TEN2 dimer, in which the LPHN site is no longer accessible. In contrast, TEN2 without a splice insert is free to make a trans-interaction with LPHN and a cis-interaction with FLRT3, thus specifying an excitatory synapse.

Overall, the new findings represent a substantive extension (and correction!) of their 2018 study, advancing our understanding of the molecular interactions that specify synapses. My enthusiasm is severely dampened, however, by the sly way in which the correction is presented: slipping a few lines of "supplementary discussion" into a supplementary figure legend (Fig. S6) won't do – the issue needs to be confronted in the main text and with additional experiments. In addition, the manuscript also suffers from being poorly organised in places. My detailed comments are as follows:

Major point 1: The substantial correction of their own previous work must be presented in a more candid manner. A soluble TEN2 Δ Tox protein should be tested for LPHN binding, and biophysical methods should be used to investigate the stability and native structure of this deletion construct (e.g. differential scanning fluorimetry, CD and NMR spectroscopy). Given that mutagenesis is used to validate the new LPHN binding site, it is important that we know what went wrong in the previous experiments.

Major point 2: The results shown in Figs 4 and 5 should be described in a more logical manner. On page 9, the "DHR" mutation is introduced, but before its effect on LPHN binding is revealed, the text moves on to a description of alternative splicing. It may better to first show the data relating to the wild-type situation +/- SS (current Fig. 5) and then the data relating to the DHR mutation (current Fig. 4). The schematics (current Fig. 4A) are very useful, but they do not need to be repeated in every panel. In Fig. 5B, the quantitation is missing.

Major point 3: The term "cis" interaction is used incorrectly to describe the interaction between a membrane-bound protein and a soluble protein. This is misleading and inconsistent with the previous correct usage in Li et al (2018). An alternative term should be used.

Major point 4: The model in Fig. 7 should be tested by mutagenesis of the salt bridge interactions that mediate TEN3 dimerisation. This would be a more stringent test of the model than comparing -SS and +SS.

Major point 5: What is the "LYR" mutation in Figs S5 and S6?

Minor point 6: In the previous paper and in many of the present figures, teneurins and latrophilins are abbreviated as TENs and LPHNs. However, in the abstract and text, they are "Tenm's" and "Lphn's". Please be consistent and/or drop the misguided apostrophes.

Minor point 7: Line 178 – Please add reference to LPHN3/FLRT3 structure.

Minor point 8: Lines 212-214 – Please use single-letter codes for amino acid residues. This looks like it was pasted from PYMOL.

Minor point 9: Replace "glycochain" with "glycan".

Minor point 10: Remove reference to "Leon et al., Net Commun., acceptor in principle" (sic). This can be added back at a later stage, if needed.

Minor point 11: Fig. 2A,B would be more logical in Fig. 1.

Reviewer #2:

Remarks to the Author:

Li et. al. report cryo-EM structures of trans-synaptic adhesive teneurin-latrophilin complex, defining binding interface between these two proteins. Moreover, based on biochemical and cell surface binding studies, the authors propose a mechanism of how a remote alternative splicing site distal to the binding interface can regulate teneurin/latrophilin complex formation and, consequently, specify inhibitory or excitatory synapse development. The proposed model is novel and would greatly help understand the roles of teneurin/latrophilin in synapse specification and development.

I have several major comments about the manuscript:

1. In supplemental Figure S4, the Lphn3 lectin domain appears to be of low local resolution and only allows for tracing main chain, but not side chain assignment essential for defining interactions at lectin/beta-barrel interface shown in Figure 3. If this is the case, I would suggest not to include specific atomic interactions in the text, or simply let readers know that these interactions are tentative.

2. Similarly, the disulfide bond assignment in the Lphn3 lectin domain (page 3) might not be supported by this map also.

3. In Panel E of Figure 2, it is not clear whether the labeled triple complex peak represents a Lphn3/Ten/FLRT complex because the FLRT band is much weaker compared to other two components and because Fraction 8 contains more FLRT than Fraction 4 or 5. I would think that Fraction 4 or 5 at the center of putative complex peak should contain more FLRT than the flanking fractions.

4. In panel E (left) of Figure 4, it would be more appropriate to show gel filtration profile of TEN ECRΔ1_DHR, instead of TEN ECRΔ1.

5. In panel C of Figure S1, it would help to also include FSC between model and map.

Minor comments:

1. In page 7, the reference for Lphn3/FLRT3 complex is missing.
2. Maybe avoid citing experimental figures in the introduction section?

Response to Reviewer Comments

Structure of the teneurin-latrophilin complex: Alternative splicing controls synapse specificity by a novel mechanism
NCOMMS-19-30844843

→Dear Dr. Mieck,

Thank you very much for the prompt handling of our time-sensitive manuscript. We are delighted that the reviewers appreciate the importance of our work for a variety of fields and thank them for their constructive comments. We have revised the manuscript accordingly and we believe that we are now presenting an improved version of our manuscript. We are especially excited about some of the new data we obtained that further support the new mechanism that we proposed. Below we provide our detailed responses:

Reviewer #1 (Remarks to the Author):

The present manuscript follows a publication by the same authors last year (Li et al, Cell, 2018). In that publication, the authors reported the structure of the TEN2 extracellular region and showed that alternative splicing within this region regulates trans-cellular adhesion to LPHN3, and consequently synapse specification. Using deletion mutants, the LPHN binding site on TEN2 was mapped to the toxin-like domain, >80 Å away from the synapse-specifying splice insert. It was suggested that "alternative splicing may facilitate a conformational change of the entire [extracellular region] that renders the LPHN binding site inaccessible to the full-length LPHN on the neighboring cell".

→ We thank the reviewer for closely following our research and agree with his/her summary of our previous results.

In the present manuscript, the authors have determined the structure of a soluble TEN2-LPHN3 complex, which shows that the LPHN3 lectin domain does not bind to the TEN2 toxin-like domain as previously reported, but to the side of the β -barrel. The new (and likely correct) binding site is closer to the splice site, but still too distant for a direct effect on LPHN binding. Using a series of elegant interaction experiments, the authors show that the splice insert in TEN2 has no effect on LPHN binding in solution, but that it abrogates binding in a trans-cellular aggregation assay. A triple mutation at the new LPHN binding site selectively abolishes excitatory (but not inhibitory) synapse formation. The findings are combined into a new model, in which the splice insert promotes the formation of a tight TEN2 dimer, in which the LPHN site is no longer accessible. In contrast, TEN2 without a splice insert is free to make a trans-interaction with LPHN and a cis-interaction with FLRT3, thus specifying an excitatory synapse.

→ We agree with the reviewer's summary of our results.

Overall, the new findings represent a substantive extension (and correction!) of their 2018 study, advancing our understanding of the molecular interactions that specify synapses. My enthusiasm is severely dampened, however, by the sly way in which the correction is presented: slipping a few lines of "supplementary discussion" into a supplementary figure legend (Fig. S6) won't do – the issue needs to be confronted in the main text and with additional experiments. In addition, the manuscript also suffers from being poorly organised in places. My detailed comments are as follows:

→ We thank the reviewer for his/her appreciation of our work, enthusiasm and constructive comments. We agree that in our previous paper we reported that the toxin domain is the binding site of latrophilin, but in our current study the structure shows a different binding site. We agree that we need to present the correction in a better way and organize the manuscript accordingly. We now confront the issue in the main text and provide additional experiments. We better organized the manuscript and addressed the reviewer's comments below:

Major point 1: The substantial correction of their own previous work must be presented in a more candid manner. A soluble TEN2 Δ Tox protein should be tested for LPHN binding, and biophysical methods should be used to investigate the stability and native structure of this deletion construct (e.g. differential scanning fluorimetry, CD and NMR spectroscopy). Given that mutagenesis is used to validate the new LPHN binding site, it is important that we know what went wrong in the previous experiments.

→ We thank the reviewer for raising this point and we apologize that this point was not clearly presented in the previous version of the manuscript. In this version, we made the below changes/experiments:

In summary:

1) We moved the supplementary discussion to the main discussion and expanded on the discussion.

2) We made the constructs and the viruses for the soluble TEN2 Δ Tox protein as the reviewer suggested. However, these constructs did not express in spite of all our efforts. As a result, we were not able to perform the proposed experiments. However, the lack of expression is consistent with our explanation that the same protein in mammalian cells was not properly folded.

3) We performed other experiments with the hope of addressing the reviewer's concerns. We added a new Supplementary Figure 7 testing the latrophilin binding ability of different domains of teneurin in size exclusion experiments. The results suggest that LPHN does not bind to the toxin domain

4) We updated Supplementary Figure 5 and 6; and provided further data testing the latrophilin binding ability of the toxin-domain. The results suggest that LPHN does not bind to the toxin domain.

5) During the revisions of our manuscript, another group published the chicken TEN2/mouse LPHN3 structure¹. Both structures agree that LPHN binds to the beta-barrel.

In length:

We apologize for our lengthy response below but we felt like it is important to provide our efforts on this important point:

As the reviewer summarizes, in our previous manuscript we concluded that toxin-domain of teneurin is the site for the interaction of full length TEN2 with latrophilin because deletion of this domain from the full length TEN2 (TEN2 Δ Tox construct) led to lack of LPHN binding in flow cytometry experiments. This construct, however, was properly trafficked to the cell surface suggesting that it has no folding problems. Our result was also consistent with a previous study that reported that the C-terminal region of teneurin is the site for latrophilin interaction². The observation that the protein is trafficked to the cell surface is an indication that the protein has no folding problems because unfolded or misfolded proteins cannot escape the protein quality control system in the cells and are targeted for degradation. Testing cell-surface expression is a commonly used experimental approach by many scientists although it is not an absolute proof of folding.

After we obtained the teneurin/latrophilin structure and saw that latrophilin binds to the barrel and not to the toxin domain, we suspected our own results and repeated all the experiments and observed identical results in the Araç lab. We also sent the same constructs to the Sudhof lab where they have used cell-surface staining experiments to test for both cell-surface expression and latrophilin binding. All results were consistent with each other and suggested that the full length TEN2 Δ Tox construct can be expressed on the mammalian cell surface, but it lacks the ability to bind latrophilin. We now think that these results altogether can be explained as: TEN2 Δ Tox construct somehow tricks the protein quality control system in the mammalian cells, escapes from being targeted to the degradation pathway and can make its way to the cell-surface. However, the absence of the toxin domain likely leads to a misfolded barrel domain (It is possible that the barrel is bent or folded-over etc) so that the latrophilin binding surface on the other side of the toxin domain is not capable of interacting with latrophilin.

The experiments we performed in detail are:

Point 2) As the reviewer suggested, we cloned TEN2 Δ Tox -SS and TEN2 Δ Tox +SS constructs, sequenced the DNA, produced baculovirus, and tested the protein expression in small scale. Unfortunately, none of the TEN2 Δ Tox constructs expressed in insect cells (Figure Xa). Because the wild type TEN2 expression is increased by coexpression with latrophilin, we tried co-expression to express TEN2 Δ Tox constructs

(Figure Xb,c). However, co-expressing the TEN2 Δ Tox constructs with Latrophilin lectin domain (Figure Xb) or with Latrophilin full ECR (Figure Xc) did not increase the expression of TEN2 Δ Tox constructs. We also tried large scale expression for a weakly expressing construct but was not able to obtain any protein and we were not able to perform differential scanning fluorimetry (that requires 25 μ l of 4 μ M protein), CD (that requires 100 μ l of 10 μ M protein) or NMR spectroscopy experiments (that requires similarly high amounts of protein) as the reviewer suggested. However, the lack of expression of the TEN2 Δ Tox proteins suggest that they are not properly folded and that the Tox domain is not necessarily the binding site for latrophilin.

Small-scale test

Figure X: TEN2 constructs that lack the toxin-like domain fail to express in insect cells. Small-scale test indicates that although the TEN ECR Δ 1 -SS construct expressed at high amounts, the TEN2 lacking the Tox-like domain constructs (TEN ECR Δ 1 Δ 5 -SS or TEN ECR Δ 1 Δ 5 +SS) cannot be expressed in High-Five insect cells. TEN ECR Δ 1 -SS protein band is indicated with a red box (a). Co-expression of TEN2 constructs either with the soluble LPHN3 lectin domain (b) or the LPHN3 ECR (c) has no effect on the expression of TEN ECR Δ 1 Δ 5 -SS or TEN ECR Δ 1 Δ 5 +SS constructs.

As a side note, although we agree that these methods are the best to investigate the structure and stability of a protein, this insect cell construct is not the same construct as is used in our previous experiments which were performed by full-length mammalian cell constructs. In addition, the insect cells that are used to express the soluble construct is different than the mammalian cells and thus, whatever result we will get, will not explain the issue with our previous construct that was expressed in mammalian cells.

With an effort to explain the issue as much as possible, we performed the below experiments:

Please note that cell-surface expression and proper protein folding is always a big problem with this very large multi-domain protein. Until now, we have cloned more than 100 constructs, most of which never fold and express on the cell surface and thus we cannot perform the most obvious experiments. For instance, the barrel domain on its own cannot be expressed in mammalian cells (as soluble protein, or fused to TM or fused to TM+EGF or fused to toxin) or in insect cells. Thus, we could not test its direct interaction with latrophilin.

Point 3) We performed gel filtration experiments to test latrophilin binding to the isolated domains of teneurins expressed in insect cells (using the viruses that we had made previously) (Supplementary Figure 7). We observed that isolated EGF repeats (domain 1), isolated Ig (domain 2), isolated propeller (domain 3), or isolated toxin domain (domain 5) does not interact with latrophilin. Isolated barrel (domain 4) or barrel+toxin (domains 4+5) cannot be expressed and thus could not be tested. These results again suggest that toxin domain is not the binding site for latrophilin.

Point 4) We expressed toxin domain fused to the TM domain (tried numerous constructs) or as GPI-anchored to the membrane in mammalian cells (Supplementary Figure 6 and 5). The best expressing constructs we could get (which still have expression problems) do not bind to latrophilin, suggesting that toxin domain is not the latrophilin binding site. (Supplementary Figure 6 and 5)

Lastly, we want to clarify that the major conclusions we report in our previous manuscript is still completely true except for the exact binding site of latrophilin. Our important conclusion that “latrophilin binding site is away from the alternatively spliced site” still stands and is critical. We regret that we wrongly reported that toxin domain is the latrophilin binding site, however, the experiments we performed was performed properly with the right controls and the conclusions were based on the experimental results. In our opinion, using cell-surface expression levels as a means of proper protein folding for membrane-anchored proteins in mammalian cells is still one of the most reliable and feasible way to test for mutants that will not even express, or fold properly (especially when we need to deal with hundreds of constructs). However, this method may occasionally lead to wrong conclusions and we regret that.

Major point 2: The results shown in Figs 4 and 5 should be described in a more logical manner. On page 9, the "DHR" mutation is introduced, but before its effect on LPHN binding is revealed, the text moves on to a description of alternative splicing. It may be better to first show the data relating to the wild-type situation +/- SS (current Fig. 5) and then the data relating to the DHR mutation (current Fig. 4).

→ Thank you for raising this point. We now briefly explain the effect of the DHR mutation on latrophilin binding and clarify that it will be explained better below. We still feel the necessity of mentioning alternative splicing and the different experimental setups in the same order because the reason why we had to design different experimental setups can only make sense after talking about alternative splicing. We think that switching the order will lead to more confusion. If the reviewer still thinks the order needs to be switched, we will be happy to do the change.

The schematics (current Fig. 4A) are very useful, but they do not need to be repeated in every panel.

→ We have now removed the schematics from unnecessary panels. However, we actually think that keeping them makes the manuscript more clear so if we are allowed, we prefer to put them back.

In Fig. 5B, the quantitation is missing.

→ We have now added the quantification for these constructs and for a new mutant (salt bridge mutant) in new Figure 7b (see below). As Fig 5b and 7b shares the same constructs as control; for Fig. 5b, we decided to use the data from our previous manuscript with permission from the journal.

Major point 3: The term "cis" interaction is used incorrectly to describe the interaction between a membrane-bound protein and a soluble protein. This is misleading and inconsistent with the previous correct usage in Li et al (2018). An alternative term should be used.

→ We have changed the term "cis" to "cis-like" and added a sentence clarifying that it is not real cis interaction.

Major point 4: The model in Fig. 7 should be tested by mutagenesis of the salt bridge interactions that mediate TEN3 dimerisation. This would be a more stringent test of the model than comparing -SS and +SS.

→ We thank the reviewer for suggesting this great experiment that will test the model. We designed TEN2 mutations on both full-length TEN2 -SS and +SS to break the two salt bridge interactions that mediate the TEN2 +SS cis-dimerization (new Figure 7a). 7E1154A, H1161A, R1183A mutations were made on the TEN2 -SS construct. On the TEN2 +SS construct, besides E1154A, H1161A, R1183A, a fourth amino acid (E)

located in the splice site (NKEFKHS) was mutated to A. We tested the cell-surface expression of the new constructs and observed no defects. We performed cell aggregation assays to see if breaking the salt bridges on TEN2 +SS will make TEN2 +SS behave like TEN2 -SS and restore Lphn-binding ability in cell-aggregation assays. We were excited to observe that the TEN2 +SS salt bridge mutant was indeed able to restore LPHN binding ability partially (new Figure 7b). This result is in support of the model we proposed. We think the reason for the partial binding is because insertion of the splice inserts provide further new interactions between the splice inserts and contribute to the rigidity of the cis-dimer. Even when the salt bridges are mutated, the TEN2 +SS is still more rigid than the TEN2 -SS and thus binds to LPHN less.

We added a new main figure (Figure 7) that shows the splice insert and the new cell aggregation experiments. The previous Figure 7 is now Figure 8.

Major point 5: What is the "LYR" mutation in Figs S5 and S6?

→ The "LR" mutation (L1990N, R1992T) on full-length TEN2 was generated with the idea of being a positive control for the "DHR" mutation. Based on the teneurin/latrophilin complex structure, the DHR mutation on the TEN2 barrel is at the binding interface between teneurin and latrophilin and breaks the teneurin/latrophilin interaction. The LR mutation is also on the TEN2 barrel but it is not at the interface and does not break the teneurin/latrophilin interaction. Both mutants are properly trafficked to the cell surface, and thus have no localization defects. The "LR" mutation was mistakenly labeled as "LYR" in Supplementary Figs 5 and 6. We have now corrected the mistake.

Minor point 6: In the previous paper and in many of the present figures, teneurins and latrophilins are abbreviated as TENs and LPHNs. However, in the abstract and text, they are "Tenm's" and "Lphn's". Please be consistent and/or drop the misguided apostrophes.

→ We have now changed all the abbreviations back to "TEN" and "LPHN" consistent with the previous paper.

Minor point 7: Line 178 – Please add reference to LPHN3/FLRT3 structure.

→ We have now included the citation in this sentence.

Minor point 8: Lines 212-214 – Please use single-letter codes for amino acid residues. This looks like it was pasted from PYMOL.

→ We have now used the single-letter codes for amino acid residues.

Minor point 9: Replace "glycochain" with "glycan".

→ We have replaced "glycochain" with "glycan".

Minor point 10: Remove reference to "Leon et al., Net Commun., acceptor in principle" (sic). This can be added back at a later stage, if needed.

→ The mentioned reference is now published and we now cite it properly.

Minor point 11: Fig. 2A,B would be more logical in Fig. 1.

→ We agree with the reviewer. Due to the already large size of Figure 1, we did not move Fig. 2A,B to Fig.1. Fig 2A,B also helps to visualize Fig. 2C and D better because the C-D panels are presented as schematic for the A-B panels. However, if the reviewer still thinks 2A,B belongs to Fig. 1, we will be happy to move it to Fig. 1.

Reviewer #2 (Remarks to the Author):

Li et. al. report cryo-EM structures of trans-synaptic adhesive teneurin-latrophilin complex, defining binding interface between these two proteins. Moreover, based on biochemical and cell surface binding studies, the authors propose a mechanism of how a remote alternative splicing site distal to the binding interface can regulate teneurin/latrophilin complex formation and, consequently, specify inhibitory or excitatory synapse development. The proposed model is novel and would greatly help understand the roles of teneurin/latrophilin in synapse specification and development.

→ We thank the reviewer for his/her appreciation of our work and view on the novelty and importance of the alternative splicing-dependent mechanism we propose for synapse specification. We also thank him/her for the constructive comments. We addressed all of his/her concerns below.

I have several major comments about the manuscript:

1. In supplemental Figure S4, the Lphn3 lectin domain appears to be of low local resolution and only allows for tracing main chain, but not side chain assignment essential for defining interactions at lectin/beta-barrel interface shown in Figure 3. If this is the case, I would suggest not to include specific atomic interactions in the text, or simply let readers know that these interactions are tentative.

→ We have included in the following text in the description of the interface: "In our map, the Lec domain was not as well resolved as TEN2 (Supplementary Figure 1D & Supplementary Figure 4C). Therefore, the crystal structure of the Lec domain was docked as a rigid body without fitting the side chains. Nevertheless, the tentative side chains of the Lec domain reveal interesting potential interactions with the TEN2." We also emphasized the fact in the methods part as well as in the legend of Figure 3.

2. Similarly, the disulfide bond assignment in the Lphn3 lectin domain (page 3) might not be supported by this map also.

→ We think the reviewer is referring to the disulfide bond descriptions on page 6. We deleted the sentence related to the disulfide bonds in the Lec domain. The crystal structure of the Lec domain was docked without changing the assignment of the disulfide bonds.

3. In Panel E of Figure 2, it is not clear whether the labeled triple complex peak represents a Lphn3/Ten/FLRT complex because the FLRT band is much weaker compared to other two components and because Fraction 8 contains more FLRT than Fraction 4 or 5. I would think that Fraction 4 or 5 at the center of putative complex peak should contain more FLRT than the flanking fractions.

→ The reviewer is right that the proteins on the gel look a bit weird. We have been observing such issues as we worked with these large proteins. The reason FLRT band is much weaker compared to other two components is because of the variable molecular weight of these proteins. Teneurin band is very thick because it is a very large 215 kDa protein, Latrophilin is about 100 kDa and FLRT is only 40 kDa. The gel is stained by Coomassie and the thickness/weakness of the bands also reflects their molecular weight. Thus, a 1:1:1 triple complex indeed is expected to look like FLRT is much weaker than TEN.

Similarly, fraction 8 contains more FLRT than fraction 4 or 5 because the FLRT in fraction 8 belongs to the binary Latrophilin/FLRT complex that eluted around 11 ml. Please note that Latrophilin band that comes from the binary complex is also thicker than the latrophilin band that comes from the trimeric complex.

The reviewer is right that we would expect to see more FLRT at the center of the peak than the flanking fractions. We do not yet have an explanation for this observation. However, we have been observing this phenomenon for these complexes very reproducibly and obtained the binary complex structure from a prep that looked similar.

4. In panel E (left) of Figure 4, it would be more appropriate to show gel filtration profile of TEN ECRΔ1_DHR, instead of TEN ECRΔ1.

→ Thank you for raising this point. We believe that the reviewer meant panel E (right, not left). We have now replaced the gel filtration profile of TEN ECRΔ1 with that of TEN ECRΔ1_DHR (which did not change the result).

5. In panel C of Figure S1, it would help to also include FSC between model and map.

→ We have included the FSC between the model and the map in Supplementary Figure 1.

Minor comments:

1. In page 7, the reference for Lphn3/FLRT3 complex is missing.

→ We added the missing reference.

2. Maybe avoid citing experimental figures in the introduction section?

→ We now avoid it as much as possible.

References

1. Del Toro D, *et al.* Structural Basis of Teneurin-Latrophilin Interaction in Repulsive Guidance of Migrating Neurons. *Cell* **180**, 323-339 e319 (2020).
2. Li J, *et al.* Structural Basis for Teneurin Function in Circuit-Wiring: A Toxin Motif at the Synapse. *Cell* **173**, 735-748 e715 (2018).

Reviewers' Comments:

Reviewer #1:

Remarks to the Author:

I am satisfied with the revision. The authors have made a good effort to investigate the source of their previous misleading result. If the required deletion mutants cannot be made as soluble proteins, there is indeed little that can be done. The new data in Fig. S7d support the notion that the toxin-like domain of TEN is not the binding site for LPHN.

The revised discussion is adequate and I commend the authors for referencing the rival study by del Toro et al. Always nice when there is agreement between two independent structures!

The other changes are also fine. I leave it to the editor to decide on figures etc.

One minor comment: in line 211, the authors say "is still capable of binding carbohydrates". It would be better to say "may still be able to bind". Continuous uninterpreted electron density is no proof of binding.

Reviewer #2:

Remarks to the Author:

I think that the recently published Cell paper confirmed the ternary complex by SPR, so I have no major concern about the manuscript.

REVIEWERS' COMMENTS:

Reviewer #1 (Remarks to the Author):

I am satisfied with the revision. The authors have made a good effort to investigate the source of their previous misleading result. If the required deletion mutants cannot be made as soluble proteins, there is indeed little that can be done. The new data in Fig. S7d support the notion that the toxin-like domain of TEN is not the binding site for LPHN.

The revised discussion is adequate and I commend the authors for referencing the rival study by del Toro et al. Always nice when there is agreement between two independent structures!

→ We thank the reviewer for his/her comments.

The other changes are also fine. I leave it to the editor to decide on figures etc.

One minor comment: in line 211, the authors say "is still capable of binding carbohydrates". It would be better to say "may still be able to bind". Continuous uninterpreted electron density is no proof of binding.

→ We have changed the sentence accordingly.

Reviewer #2 (Remarks to the Author):

I think that the recently published Cell paper confirmed the ternary complex by SPR, so I have no major concern about the manuscript.

→ We thank the reviewer for his/her appreciation of our work and the support for its publication in Nature Communication.